# A unified approach to dissecting biphasic responses in cell signaling

**Vaidhiswaran Ramesh[1], J Krishnan[1,2]\***

[1]Department of Chemical Engineering, Sargent Centre for Process Systems Engineering, Imperial College London, London, United Kingdom; [2]Institute for Systems and Synthetic Biology, Imperial College London, South Kensington Campus, London, United Kingdom

**Abstract** Biphasic responses are encountered at all levels in biological systems. At the cellular level, biphasic dose-responses are widely encountered in cell signaling and post-translational modification systems and represent safeguards against overactivation or overexpression of species. In this paper, we provide a unified theoretical synthesis of biphasic responses in cell signaling systems, by assessing signaling systems ranging from basic biochemical building blocks to canonical network structures to well-characterized exemplars on one hand, and examining different types of doses on the other. By using analytical and computational approaches applied to a range of systems across levels (described by broadly employed models), we reveal (i) design principles enabling the presence of biphasic responses, including in almost all instances, an explicit characterization of the parameter space (ii) structural factors which preclude the possibility of biphasic responses (iii) different combinations of the presence or absence of enzyme-biphasic and substrate-biphasic responses, representing safeguards against overactivation and overexpression, respectively (iv) the possibility of broadly robust biphasic responses (v) the complete alteration of signaling behavior in a network due to biphasic interactions between species (biphasic regulation) (vi) the propensity of different co-existing biphasic responses in the Erk signaling network. These results both individually and in totality have a number of important consequences for systems and synthetic biology.

**\*For correspondence:**
j.krishnan@imperial.ac.uk

**Competing interest:** The authors declare that no competing interests exist.

## Editor's evaluation

This study presents a useful mathematical analysis of different signaling networks in an attempt to provide general rules that give rise to biphasic responses, a widely observed behavior in biology in which the outputs of the network depend non-monotonically on the inputs. The methodology is comprehensive and solid, and should provide a useful starting point for systems biologists and quantitative biologists interested in engineering synthetic biological systems and for mechanistically understanding biphasic responses in natural biological systems.

## Introduction

The enzymatic modification of substrates is the basic building block of cell signaling and cellular biochemical networks (*Martin, 2014*; *Ventura et al., 2010*; *Conradi and Shiu, 2018*). The basic intuition from the analysis of these systems is that increasing both the concentration of the enzyme performing the modification, and total concentration of the substrate increases the steady state concentration of the modified substrate.

However, observing both pathways and networks which are comprised of these building blocks indicates that there may be significant divergence from this basic behavior. One of the most widespread ways in which this may manifest is through a biphasic dose response (*Levchenko et al., 2000*;

*Suwanmajo and Krishnan, 2013*; *Witzel and Blüthgen, 2018*; *Kamenz et al., 2021*; *Szomolay and Shahrezaei, 2012*). Far from being an accidental occurrence, biphasic dose responses have many advantages, including a built-in attenuation mechanism, a cap on the maximal level of output as well as the presence of a zone of 'optimal' activation. This is borne out by the fact that these very same implications have important consequences in different contexts.

Biphasic responses (both steady state dose responses and dynamic responses) have been recorded all across the landscape of cellular signaling including in key signaling pathways such as Ras, PI3K/ Akt, MAPK/Erk, p53 (*Shen et al., 2020*; *Kortholt et al., 2013*; *Zhu et al., 2011*; *Zhou et al., 2011*; *Zhuang et al., 2020*; *Heltberg et al., 2019*; *Neumann et al., 2020*; *Park et al., 2002*; *Reising et al., 2022*; *Kanodia et al., 2014*), and gene regulation (*Wong et al., 2011*; *Kim et al., 2012*; *Wang et al., 2020*; *Liu et al., 2021*), arising from a number of factors including specific network motifs (*Kim et al., 2008*) and protein structural factors (*Karanicolas and Brooks, 2003*). Biphasic responses also leads to biphasic regulation in networks. As examples of biphasic regulation in networks, protein modification of myosin light chain (phosphorylation) is regulated biphasically by p21-activated kinase (*Chu et al., 2013*) Lysophosphatidylcholine, can induce biphasic regulation of NF-kB activity through a PKC dependent pathway in human vascular endothelial cells (*Sugiyama et al., 1998*). Sometimes, the biphasic dose-response behavior is only present under certain conditions or in the presence of a different protein. For example, corticosteroids have been shown to biphasically regulate surfactant protein A mRNAs in human lung cells, in the presence of bibutyryl cAMP (*Fonseca et al., 2015*). Tyrosine phosphorylation of STAT3 protein, in the presence of SOCS3, is biphasic in response to continuous IL-6 signaling, and monophasic without SOCS3 (*Wormald et al., 2006*). A recent study focusing on PKA and Erk regulation has identified how they both vary biphasically to cAMP levels (with opposite phases) (*Zara et al., 2020*). Furthermore, multiphasic responses have been observed experimentally in other contexts in plant biology (*Cvrčková et al., 2015*). Biphasic responses have been documented and discussed in a wide-array of contexts under the umbrella of hormesis (*Calabrese and Baldwin, 2003*).

Biphasic responses in signaling are often desired, and can serve as protection against overactivation, overexpression and mutant takeover (*Karin and Alon, 2017*; *Witzel and Blüthgen, 2018*). In other instances, this behavior is undesired and can result in inadequate and/or unexpected signaling (*Abbasi et al., 2016*; *Calabrese and Baldwin, 2003*). All in all, given the diverse contexts in which it is encountered, the different ingredients which give rise to it, and the diverse perspectives from which it is studied and is of interest (natural/engineered biology, desired/undesired behavior, key behavioral signature), a broad-based systems study of this behavior, and the ingredients either enabling or precluding it, is vital.

A typical way in which biphasic responses in cell signaling networks are studied is to investigate network motifs which generate such behavior (*Kim et al., 2008*; *Yang and Iglesias, 2006*; *Varusai et al., 2015*; *Ramesh et al., 2023*). In this paper, by contrast, we investigate the sources of biphasic dose responses, by starting with the basic building blocks of substrate modification, which represent the most basic elements of signaling and post-translational modification networks. By investigating a series of basic building blocks, we reveal the potential for basic modifications to intrinsically give rise to biphasic dose responses, either in response to enzyme concentration or substrate concentration variation. In so doing we obtain a unified synthesis of biphasic responses in response to both enzyme and substrate variation (which could represent safeguards against protein overactivation and overexpression, respectively). We then focus on the network level, building on these results to investigate how these sources of biphasic responses are intrinsic to chemical modification affect network behavior. The study is rounded out by an analysis of a concrete exemplar system (Erk signaling), leading to a re-evaluation of existing hypotheses and assumptions complemented by a unified synthesis of biphasic responses here.

Biphasic responses are encountered at all levels in biological systems. In this paper, we address the question: what types of biphasic responses are encountered in cell signaling, what are the factors which contribute to this widely observed behavior and what are its consequences? Our unified analysis reveals that the propensity for different types of biphasic responses (with respect to signal, and with respect to protein amounts) is woven into the fabric of cellular biochemical networks, and has a number of non-trivial implications for overall network behavior.

## Results

From a dose-response perspective, enzyme-mediated protein (substrate) modification has a natural response (concentration of the modified substrate) and two natural doses: (1) The total enzyme amount, and (2) The total substrate amount. In this paper, using the exemplar of phosphorylation for such protein modifications, we study the capacity of networks and systems involving enzyme-mediated protein modification to show biphasic dose responses in the concentration of the (maximally) modified substrate form with each of these doses. The biphasic dose responses in the (maximally) modified substrate with increasing total amounts of substrate and enzyme are henceforth called substrate biphasic and enzyme biphasic responses, respectively. In each case, biphasic responses arise as a consequence of competing effects, which emerge from increasing the relevant dose, and this is discussed further below.

The organization of the results reflects the multi-pronged nature of our analysis (*Figure 1*). We first focus on the biochemical modification level and analyze the capacity of simple modules of enzymatic substrate modification (such as the covalent modification cycle and basic extensions thereof) to exhibit biphasic dose responses. Following this, we focus on the network level and analyze the consequences of such dose responses for network behavior by analyzing simple network motifs with biphasic interactions. Finally, we dissect different aspects of biphasic responses in a specific signaling system: ErK regulation, which is a well characterized cellular pathway shown to present biphasic dose response behavior (*Aoki et al., 2013*).

Our results involve a combination of direct demonstration of the presence of biphasic responses as well as an unambiguous ruling out of biphasic responses for different systems and different doses. The presence of a biphasic response is demonstrated computationally by performing a one-parameter bifurcation analysis (varying the relevant dose: total amounts of substrates or enzymes). In each of the cases studied, we use analytical work (see models and methods) to ascertain the presence or absence of biphasic behavior in the model.

Analytical work is used to reveal the conditions (intrinsic kinetic parameters and total amounts of substrates and enzymes) under which a given system can exhibit biphasic dose responses. In other cases, it is used to show how the behavior is impossible in the system for any choice of kinetic parameters and total species amounts. A discussion of the different classes of parameters (intrinsic kinetic parameters, total species amounts) is provided in the models and methods. We elaborate on a few aspects below.

As part of our analysis of biphasic responses, we aimed to assess the extent to which underlying parameters can prevent or enable the presence of biphasic responses (to either type of dose). We first note that the parameters are of two types: intrinsic kinetic rate constants (the majority of the parameters) and the total amounts of enzymes(s) and/or substrate (other than the dose). The main challenge which arises is the fact that there are a number of intrinsic kinetic parameters (which do not cluster into a small number of groups of parameters, like in many physical systems). From our analysis, we find, especially in studies at the biochemical modification level, three kinds of scenarios: (a) Biphasic responses are impossible for any values of the intrinsic kinetic parameters, thus categorically ruling out their possibility, from structural considerations. (b) Biphasic responses are possible for certain regions of intrinsic kinetic parameter space, which are explicitly characterized (these are necessary conditions). In these cases, we can guarantee the presence of a biphasic response, for suitable values (and ranges) of total amounts of enzymes and substrates. Thus, here, there is a partial restriction on the intrinsic kinetic parameter space for enabling the possibility of a biphasic response. (c) Biphasic responses are possible irrespective of the intrinsic kinetic parameter space. Thus intrinsic kinetic parameters play no essential role in restricting the possibility of a biphasic response (and thus biphasic responses are a widespread occurrence in the space of intrinsic kinetic parameters). Furthermore, a biphasic response can be guaranteed for suitable ranges of enzyme and substrate total concentrations.

All in all, our approach presents a unified analysis of biphasic responses in signaling systems, by assessing both enzyme and substrate biphasic responses, examining signaling networks from three complementary perspectives (biochemical building blocks, network topologies, concrete exemplars) and in all cases uses analytical and computational work to obtain both structural and parameter-dependent insights.

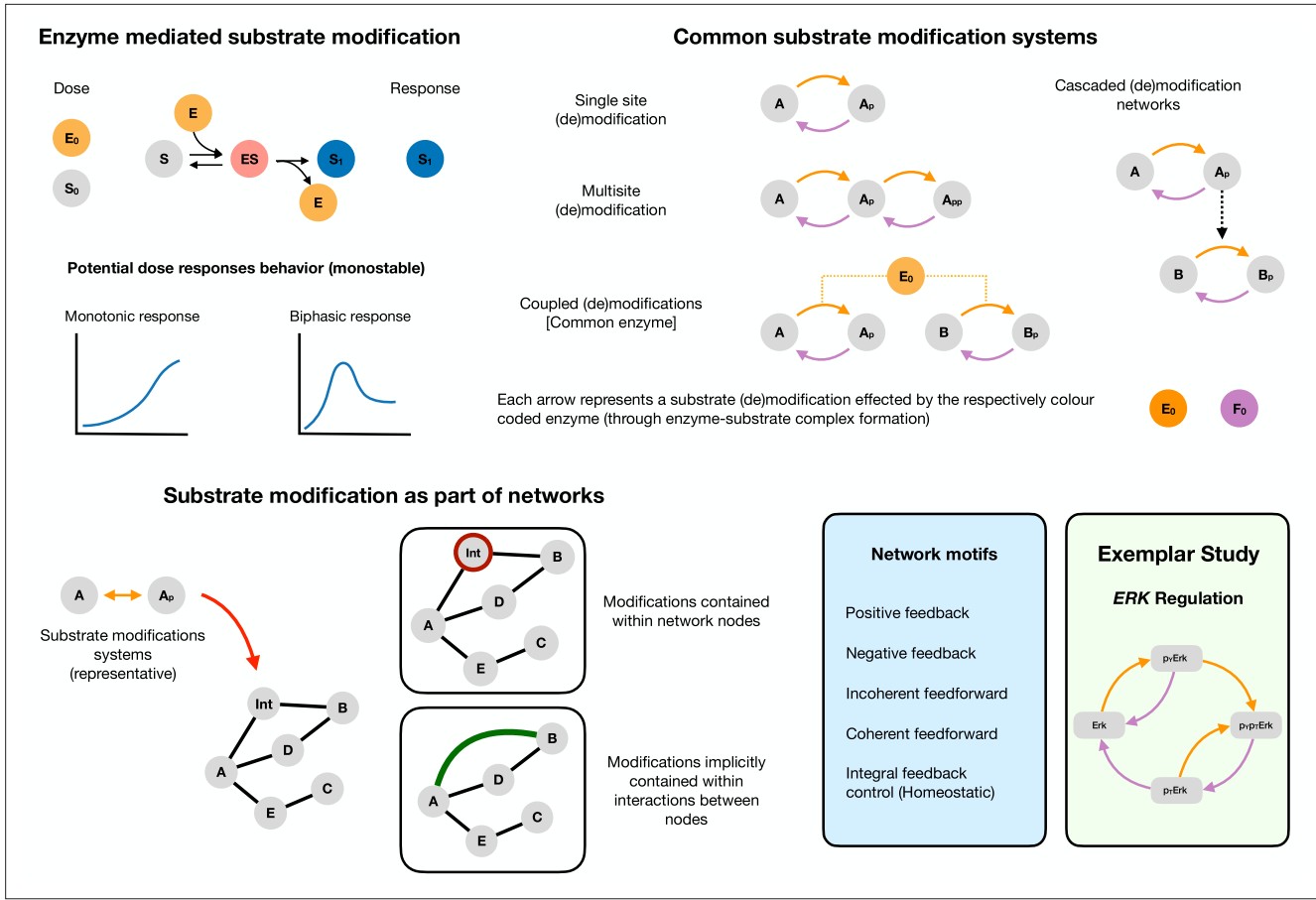

**Figure 1.** Biphasic dose responses in basic building blocks in cell signaling: from enzyme-mediated substrate modification systems to network motifs/modules Reversible enzyme-mediated substrate modification is an integral part of cellular signaling and is among its basic building blocks. *Top left* shows the basic mechanism of substrate modification by an enzyme involving the following steps: the binding (and unbinding) of the substrate ($S$) with the enzyme ($E$) to form a substrate-enzyme complex ($ES$), which finally disassociates to give the modified substrate ($S_1$) and the free enzyme ($E$). The reverse modification is not depicted for conciseness. The output of the system is the concentration of the modified substrate $S_1$. There are two natural doses associated with this - total substrate amount and total enzyme amount ($S_0$ and $E_0$, respectively). The potential dose-response behavior in the simplest instance can thus be either (**a**) monotonic or (**b**) exhibit biphasic dependence of the modified substrate with the dose. Both substrate and enzyme doses can potentially elicit biphasic responses. *Top right* shows the schematics of commonly encountered substrate modification building block systems in cell signaling. Each arrow denotes a modification and involves binding, unbinding and catalytic action by the enzyme. Arrows of the same color indicate action by the same enzyme type (a kinase and a phosphatase for example). *Bottom left* These systems form key components of wider signaling networks in the cell. From a network perspective they can be incorporated in two ways. (1) They can either be explicitly described, for example as part of a node or (2) be implicitly present within an interaction between nodes (implicit in an arrow, which can be the sum of actions of species not explicitly defined). A fruitful way to study signaling networks is through the study of recurring signaling motifs (with characteristic input-output responses) such as feedback and feedforward structures. *Bottom right* lists canonical network motifs (positive and negative feedback, and coherent and incoherent feedforward, integral feedback control) that we study in our work. We complement our study of biphasic responses in biochemical building-block systems and network motifs by studying biphasic responses in a concrete exemplar system, the *ErK* signaling pathway. Detailed mathematical model descriptions of all the systems studied are discussed in the supplementary material, and a detailed reaction schematic of the models considered is shown in *Figure 1—figure supplement 1*.

The online version of this article includes the following figure supplement(s) for figure 1:

**Figure supplement 1.** Schematic representation of the various models considered in the manuscript.

## The modification biochemistry level

Our analysis begins with the modification biochemistry level. Biphasic responses at the modification biochemistry level, emerge from competing effects engendered by sequestration of enzymes/substrates (which in turn is simply a consequence of complex formation in any enzymatic biochemical reaction).

We explore a range of biochemical modification systems which bring to light exactly when such sequestration effects will provide a competing effect giving rise to a biphasic response. We start with the simplest modification system (a covalent modification cycle), and increase the complexity systematically, allowing for multiple enzymes and/or substrates and sharing of enzymes and substrates. As seen below, these provide the ingredients for generating competing effects leading to biphasic responses. Note that in all these instances, if the sequestration effects are removed (e.g. all substrate modification steps in the unsaturated limit), the possibility of a biphasic response is eliminated.

Our analysis at the substrate modification level is underpinned by an exploration of a broad suite of basic systems, which represent different extensions of the covalent modification cycle (the basic unit of reversible substrate modification). Thus we consider (i) the covalent modification cycle, along with additional interactions (ii) multiple modifications of a substrate, mediated by either common/separate enzymes (iii) enzymatic modifications cascades (where the modified substrate at one stage of the cascade is the enzyme for the next), again considering both common and separate phosphatases. (iv) two different covalent modification cycles whose conversion is mediated by common enzymes (either kinase or phosphatase or both). Thus, this suite of basic systems allows us to thoroughly explore the impact of the sharing of enzymes or substrates, or both, in enabling (or precluding) the possibility of biphasic responses. In particular, the structural requirements for biphasic responses which emerge allow us to pinpoint key drivers and minimum requirements for such responses providing valuable mechanistic insight in the process. In all the cases studied, we employ broadly-used models explicitly describing the elementary steps of enzyme-substrate modification (binding, unbinding, catalytic conversion) and make no a priori assumptions on kinetic regimes.

## The organization of the results

We present the results of the analysis of each of these systems, in sequence. In each case (a) we indicate whether or not a biphasic response (of either type) is possible (b) we specify whether there are any restrictions in intrinsic kinetic parameters for this to happen (this is based on analytical work) (c) in systems where biphasic responses are possible, we present the basic mechanistic requirements and intuition emerging from the analysis. (a) and (b) are also depicted in *Figure 2*: here when biphasic responses are possible, a sample biphasic response is shown. We present a summary of additional parametric analysis for the systems which can exhibit biphasic responses, at the end of this subsection.

## The covalent modification cycle with additional complex formation is capable of enzyme biphasic responses

The simplest enzymatic protein modification system, the covalent modification cycle (model M0) is incapable of exhibiting biphasic responses with either increasing total enzyme amounts or total substrate amounts (see *Supplementary file 1*, section 1). This is true irrespective of model parameters. However, an additional complex formation of the modified substrate with the kinase enzyme (model M1), allows for the network to exhibit enzyme biphasic responses (see *Figure 2B*). Some aspects of the capacity of this system to exhibit enzyme biphasic dose responses have been studied earlier (albeit with different modeling assumptions by *Varusai et al., 2015*) and also as part of analyzing multispecific interactions (*Seaton and Krishnan, 2012*). Interestingly, we find that enzyme biphasic responses may be obtained for any choice of intrinsic kinetic parameters. However, this system is incapable of exhibiting substrate biphasic responses (see *Supplementary file 1*, section 1).

Examining the two cases above illuminates why an enzyme-biphasic response emerges in the later case. The extra protein interaction creates a competing effect: while an increase in kinase enzyme concentration, generally favors increased concentrations of the modified substrate, it also has a secondary effect of binding to the modified substrate and sequestering it, thus reducing the (free) modified substrate concentration. This creates the in-built competing effect resulting in a biphasic response.

## Multisite substrate modification and commonality of enzyme action promotes biphasic dose responses

Proteins rarely undergo just one (de)modification and often are (de)modified at different sites (effected by either the same enzyme or by different enzymes). To explore the consequence of such multiplicities

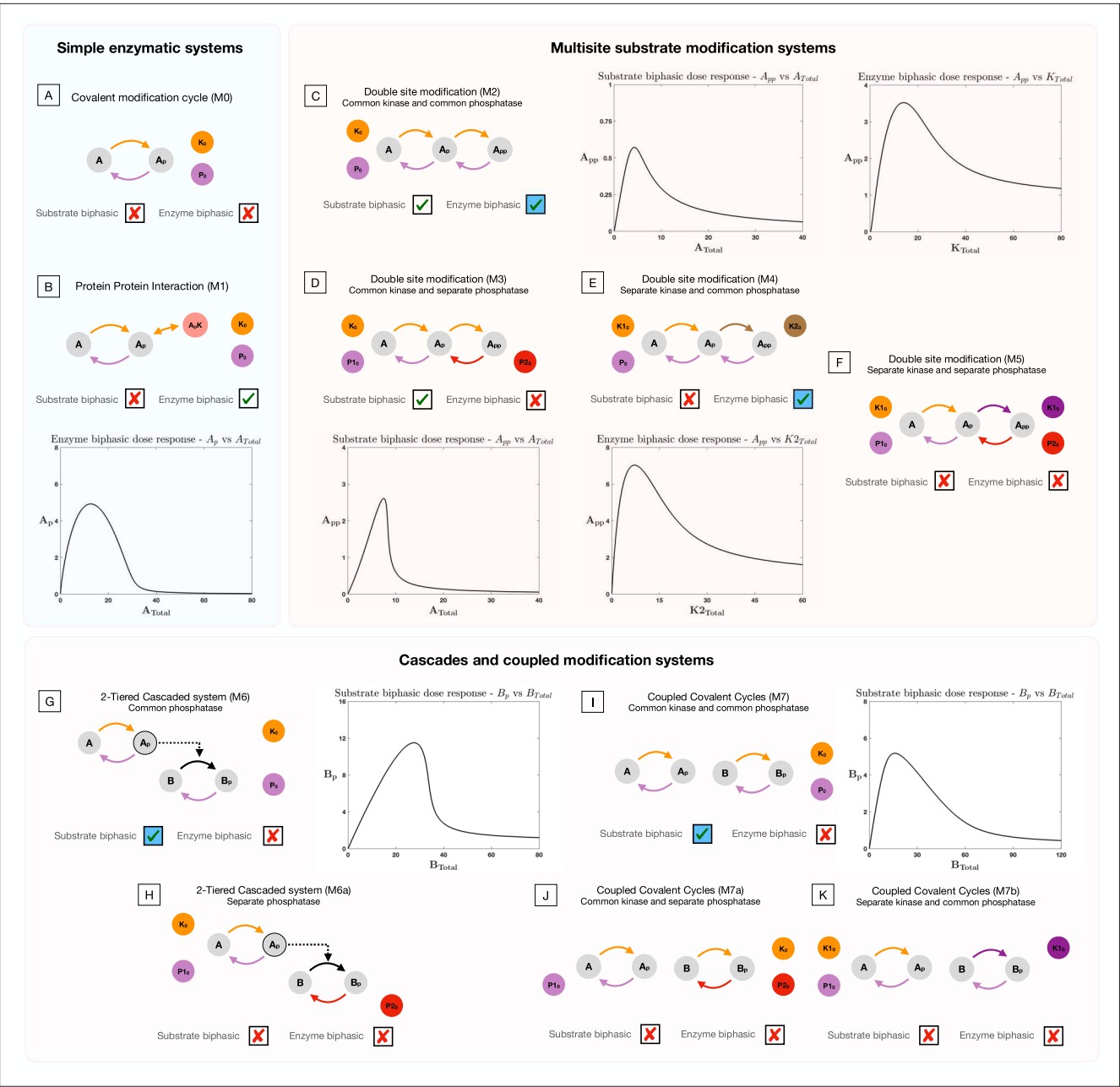

**Figure 2.** Presence and absence of substrate and enzyme biphasic dose responses in the commonly observed building blocks of cellular signaling systems. An examination of a suite of substrate modification systems and different doses allows us to clearly ascertain the origin and necessary features of such systems to present enzyme and substrate biphasic dose responses. Where results denote a biphasic response is absent (indicated by a cross in a box), this is absent irrespective of kinetic parameter values and total amounts of substrate(s) and enzyme(s). This is established through analytical work (see *Supplementary file 1*). When a biphasic dose-response is present, it is shown in a bifurcation diagram where the relevant dose is the bifurcation parameter. The presence of specific dose-responses can be characterized in parameter space in the following way, either the behavior is (1) present for *all intrinsic kinetic parameter values* (accessible for some total amounts of substrate and enzyme, transparent boxes with a tick), or (2) present only for *specific intrinsic kinetic parameter values* (accessible for some total amounts of substrate and enzymes, blue-shaded boxes with a tick) - see text and analytical work for more details. (**A**) *Covalent modification system.* Absence of substrate and enzyme biphasic response. (**B**) *Protein-Protein Interaction Model.* Enzyme biphasic dose responses are seen in the protein-protein interaction model, but substrate biphasic responses are absent. In contrast to the covalent modification cycle which is incapable of biphasic responses, this result indicates how a single additional complex formation by the enzyme (and resulting sequestration) can generate enzyme biphasic dose responses. (**C**) *Double site modification (DSP): common enzymes.* Presence of enzyme and substrate biphasic response (enzyme biphasic responses only for certain ranges of intrinsic kinetic parameter values - see text). (**D**) *DSP: common kinase and separate phosphatase.* Presence and absence of substrate and enzyme biphasic response, respectively. (**E**) *DSP: separate kinase and common phosphatase.* Absence and presence of substrate and enzyme biphasic response, respectively (enzyme biphasic only with respect

*Figure 2 continued on next page*

*Figure 2 continued*

to the total amount of the second kinase, and only for certain ranges of intrinsic parameter values- see text). (**F**) *DSP: separate kinase and separate phosphatase*. Absence of both substrate and enzyme biphasic response. These results together show how commonality of both enzymes promotes biphasic responses with both doses, and in particular how commonality in phosphatase and kinase action enables enzyme and substrate biphasic responses, respectively. (**G, H**) *Two-tier enzymatic cascades with common and separate phosphatases, respectively*. Presence of substrate biphasic in the second-tier substrate of the two-tier cascaded enzymatic modification system with a common phosphatase. The first tier substrate is incapable of substrate biphasic in the same model. Both tier substrates are incapable of substrate biphasic responses when the phosphatases are distinct. Enzyme biphasic responses are absent in both systems. (**I–K**) *Coupled covalent modification cycles with common and separate enzymes*. Presence and absence respectively of substrate and enzyme biphasic responses in the system with common kinases and common phosphatases. The system with separate phosphatases and the system with separate kinases are incapable of substrate and enzyme biphasic responses. (**G–K**) indicates how commonality of enzymes can enable biphasic responses in covalent modification systems, which are either otherwise decoupled (**I, J**) or where they are part of a cascade (**G, H**). This highlights how such features emerge, even though the constituent modules (covalent modification cycles) are incapable of it.

The online version of this article includes the following figure supplement(s) for figure 2:

**Figure supplement 1.** Coexistence of multi-stability and biphasic dose response in various enzymatic models for the same underlying kinetic regime.

**Figure supplement 2.** Biphasic responses in random double site modification network with separate enzymes effecting each modification.

**Figure supplement 3.** Exploration of the effect of species total amounts in allowing for biphasic responses in different modification systems.

**Figure supplement 4.** Semi-analytical approach for determining total amounts of species for realizing biphasic responses.

---

in modifications, we consider the simplest extension of a basic covalent modification cycle, with a substrate undergoing two modifications in an ordered fashion, where the order of phosphorylation is opposite that of dephosphorylation and the (de)modifications effected by the same or different enzymes (see models M2-M5 in *Figure 1—figure supplement 1*). In the instance where a common enzyme effects multiple modifications, the mechanism is assumed to be distributive.

We observe that the double site modification (DSP) network with common kinase and common phosphatase effecting the modifications (Model M2) is capable of exhibiting both substrate and enzyme biphasic responses in the maximally modified substrate, $A_{pp}$ (see *Figure 2C*; *Suwanmajo and Krishnan, 2013*). Our analytical work reveals the parametric dependence for this behavior to be observed. Substrate biphasic responses are guaranteed to be observed for some total amount of kinase and phosphatase for any set of intrinsic kinetic parameters of the system. On the other hand, enzyme biphasic responses can only be present when the following condition ($k_1 k_3 - k_2 k_4 > 0$) is satisfied by the kinetic catalytic constants ($k_1, k_2$ associated with phosphorylation, $k_3, k_4$ associated with dephosphorylation) of the system. This condition is a necessary and sufficient condition, and when it is violated the system is incapable of exhibiting enzyme biphasic responses (see *Supplementary file 1*, section 2).

It is interesting to note that this kinetic condition involving the catalytic constants is also observed elsewhere in studies relating to the presence of multistability in the DSP with common enzymes. *Conradi and Mincheva, 2014* identified that if the condition is violated, then the network is guaranteed to exhibit multistability at some total amounts of substrate and enzyme. These results together suggest that irrespective of the kinetic regime of the system, the DSP with common enzymes is capable of exhibiting either multistability or biphasic dose response with enzyme necessarily (for some total amounts of substrates and enzymes). This reveals the plurality of information processing behaviors capable with substrate modification systems with only two modification sites, and how catalytic constants (and more broadly intrinsic kinetic constants) can allow us to guarantee and generate these behaviors in such systems. It is also a striking example of how one among a suite of non-trivial behaviors is guaranteed to be observed across intrinsic kinetic parameter space.

## Additional specificity of enzymes in DSP reduces the capability of biphasic responses

An additional source of complexity in enzyme-mediated substrate modification arises from the specificity of enzyme action. For instance, the same enzyme could effect modifications on different proteins. Conversely, in multisite modification of the same protein, there could be distinct enzymes effecting each (de)modification. Considering such scenarios with different enzymes effecting each (de)modification in double-site modification (see models M3-M5 in *Figure 2—figure supplement 2*), we notice that the capacity of the DSP system to exhibit biphasic dose response progressively disappears,

with increasing specificity in enzyme action. The DSP with separate kinases and a common phosphatase (model M4) acting on each modification site is capable of only exhibiting enzyme biphasic responses, and only with the variation of the total amount of the kinase effecting the second modification (see *Figure 2E*). Associated kinetic constraints are discussed in *Supplementary file 1*, section 3. On the other hand, it is incapable of exhibiting substrate biphasic dose response irrespective of kinetic parameters. Similarly, the DSP with a common kinases and different phosphatases effecting each modification (model M3) is capable of exhibiting substrate biphasic responses (see *Figure 2D*), while it is incapable of exhibiting enzyme biphasic responses with changing total kinase amount. Analytical work further reveals that substrate biphasic responses in the model is guaranteed to be observed for any choice of underlying kinetic parameter choice, for some total amounts of kinase and phosphatase. The DSP with separate kinases and phosphatases effecting (de)modifications (model M5) is incapable of exhibiting either substrate or enzyme biphasic (see *Supplementary file 1*, section 3 for the relevant proofs).

An examination of all these cases reveals the factors responsible for generating the biphasic response (also see *Suwanmajo and Krishnan, 2013*). First, the possibility of an enzyme biphasic response can be explained as follows: in the common-kinase common-phosphatase case, while increasing the kinase concentration favors an increase in the fully modified substrate, it can also have an auxiliary effect: sequestering the partially modified substrate in a kinase complex. This has the effect of making more phosphatase available to dephosphorylate the fully modified substrate and this provides the in-built competing effect. The crucial factor here is the fact that the phosphatase dephosphorylates both fully modified and partially modified substrate. This is further borne out by the fact that even the separate kinase common phosphatase case can result in an enzyme biphasic response when the concentration of the second kinase is varied (for the same reason).

With regard to substrate biphasic responses, we find that the common kinase separate phosphatase case shows this. Here, increasing the total substrate concentration has two effects: the natural effect of increasing all substrate concentrations including that of the fully modified substrate, and another effect: that of reducing the free enzyme concentration. This, when combined with co-operativity, the effect of having two substrates reduce the free kinase concentration (the unmodified and partially modified form) provides a sufficiently strong nonlinear effect, allowing for a competing effect resulting in the biphasic response (confirmed by analytical work). Naturally, this effect is present in the common kinase common phosphatase case as well.

Taken together, this shows how complexity at the level of the biochemistry of a node in a signaling network (arising from the number of modifications, and enzyme sharing) can enable biphasic dose responses in the (maximally modified) substrate form. Sequestration and the commonality of enzymes here are key features that promote biphasic dose response (both enzyme and substrate).

## Enzymatic modification cascades and pathways

Modifications provide additional functionality to proteins, and in many instances, a protein undergoing a modification can gain functionality that allows it to effect modifications on another protein (*Millar et al., 2019*). This generates an enzymatic modification cascade structure (such as the MAPK system [*Kocieniewski et al., 2012*; *Kõivomägi et al., 2011*; *O'Shaughnessy et al., 2011*; *Markevich et al., 2004*]). Here too, common/separate enzymes may effect the demodifications of each of the substrates.

To explore the possibility of biphasic responses in such enzymatic modification cascades, we studied the two-tier enzymatic cascade with the demodifications being effected by either common or separate enzymes (M6 and M6a, respectively, see *Figure 1—figure supplement 1*).

This enzymatic cascade with separate phosphatases is incapable of exhibiting either substrate biphasic responses (in response to the variation in total substrate amounts in either tier) or enzyme biphasic responses in response to change in total amounts of kinase. On the other hand, while the analogous system with common phosphatases effecting demodifications on each substrate is still incapable of enzyme biphasic responses or substrate biphasic responses with respect to the first tier substrate (see *Supplementary file 1*, section 4), it is capable of exhibiting substrate biphasic responses in the second tier substrate (see *Figure 2G*). Associated kinetic constraints are discussed in *Supplementary file 1*.

This latter case can be understood as follows: increasing the total substrate concentration in the second tier has the effect of increasing the sequestering of the modified species in the first tier (in a complex with the unmodified species of the second tier). This has the consequence of making more phosphatase available, which can then dephosphorylate the modified species in the second tier. This competing effect (reminiscent of that responsible for an enzyme biphasic in multisite substrate modification above) allows for the manifestation of the substrate biphasic response.

Other forms of coupling are possible with enzymatic modifications. For example, a single enzyme can effect modifications on multiple substrates allowing for a sophisticated degree of cross-talk and coupling between pathways (*Allende and Allende, 1995*; *Ghomlaghi et al., 2021*; *Shockley et al., 2019*; *Seaton and Krishnan, 2011*). We studied the propensity for biphasic responses in the system with enzyme sharing between two covalent modification cycles. The system with a common kinase and common phosphatase for the cycles (model M7) can exhibit substrate biphasic responses in one (but not both) of the substrate modification cycles (individually, see *Figure 2I*). The system is, however, incapable of exhibiting enzyme biphasic dose responses. Analytical work further reveals that the coupled covalent modification cycles are necessarily guaranteed to present substrate biphasic responses in one of the modification cycles (but not both) for some total amount of kinase, phosphatase, and (other) substrate amount. The identity of the modification cycle which is capable of substrate biphasic responses is determined by an explicit analytical criterion which depends on intrinsic kinetic parameters.

The same system with separate enzymes acting on the two modification cycles (either kinase or phosphatase-model M7a & M7b, respectively) is incapable of exhibiting substrate biphasic responses, showing how enzyme sharing and simultaneous coupling through both enzymes is required for realizing substrate biphasic responses(see *Supplementary file 1*, section 4 for relevant proofs).

The fact that covalent modification cycles involving a shared kinase and a shared phosphatase can result in a substrate biphasic response (for one of the substrates) can be understood as follows. Increasing one of the substrates can under certain situations result in a combination of effects: one the reduction of free kinase (as seen previously) and the other an increase in free phosphatase (since the reduction in free kinase implies a lower level of modification of the other substrate, and this implies less phosphatase used there). Both these effects have been seen in different guises previously, but here their combined co-operative effect provides the nonlinearity to create the competing effect.

## Further parametric analysis

The analysis above for all systems has demonstrated either the presence or absence of biphasic responses, and analyzed to what extent the intrinsic kinetic parameters provide a fundamental restriction to the possibility of biphasic responses occurring. We now build on that to provide further insights. (1) We first note that when a biphasic response occurs, the role of intrinsic kinetic parameters are either (a) they provide no essential restriction by themselves or (b) there is a region of parameter space, explicitly delineated, which is the region where a biphasic response is possible. (2) In the later case, in all instances, there is a simple inequality involving relative magnitudes of the ratio of phosphorylation to dephosphorylation rate constants in different modifications which determines whether biphasic responses are possible or not. Equivalently, the inequality can be written in terms of relative magnitudes of products of specific catalytic constants. This emerges from detailed analysis and reflects the fact that when such a condition is satisfied, the possibility of an in-built competing effect sufficiently strong to be capable of giving rise to a biphasic response exists. (3) The type of this inequality is such that multiplying all catalytic constants by a specific constant does not affect the inequality. Thus, this inequality does not set a restriction on the absolute levels of these rate constants. Furthermore, it is clear from the nature of the inequality, that experimentally feasible ranges of kinetic parameters (catalytic constants) can satisfy such an inequality and this inequality represents a broad range of parameter space. (4) We now examine the effect of other parameters (total amounts of enzymes or substrates which are not part of the dose). By proceeding with further analytical and computational work, we can, in some cases, show further restrictions of enzyme or substrate amounts to realize a biphasic response. For instance, we show (see Supplementary maple document 2 for the points below) that higher phosphatase amounts favors enzyme biphasic responses in the DSP common kinase/phosphatase system, and also substrate biphasic in the two-tier cascade studied above. (5) Similarly we show that in both the DSP common kinase/common phosphatase (enzyme biphasic) and in the coupled

covalent modification cycles (substrate biphasic), there is a minimum level of concentration of the modified substrate to allow for a biphasic response (this incidentally also places a lower bound on the total substrate concentration for this to happen). In the coupled covalent modification cycle we can also determine which cycle will exhibit the biphasic response. (6) Further, parametric exploration into the effects of total amounts and intrinsic kinetic parameters was performed semi-analytically, starting with the analytical equations and specifying some parameters. This then provides equations for the remaining parameters and total amounts for a biphasic response to be obtained, and this was explored for all models. As part of this analysis, we also completely fixed intrinsic kinetic parameters in ranges obtained experimentally and studied computationally in *Witzel and Blüthgen, 2018* and then used semi-analytical approaches to determine the sets of two total amounts of species (enzymes or substrates) where biphasic responses are obtained (one of the set of three total amounts is the dose, this leaves, two other total amounts in the most basic case where there is a single kinase and a single phosphatase: in the case of multiple kinase and/or multiple phosphatases, there are more total amounts, and these total amounts could be explored together, or by having some of them fixed at particular values). (7) We performed semi-analytical and computational explorations for parameters sets for different modifications in the range of those experimentally determined in the MEK system, and explored in *Witzel and Blüthgen, 2018*: this is summarized in *Figure 2—figure supplement 3* and *Figure 2—figure supplement 4*. We find that biphasic responses can be obtained in ranges of intrinsic kinetic parameters, in a very similar range as that obtained there. In each model, we were able to obtain biphasic responses in reasonable physiological ranges of total amounts (and for total amounts which could be employed in synthetic settings). We also explored a second parameter set where the unbinding rate constants were set to zero, and again biphasic responses were obtained in reasonable ranges (results not shown). (8) Building on these semi-analytical and computational analyses and using continuity arguments, we can see that biphasic response can be readily obtained in neighboring regions in parameter space (both intrinsic kinetic parameters and total amounts). All in all, this complements our analysis of the intrinsic kinetic rate constants in restricting the possibility of biphasic responses (presented earlier), providing additional insights and parametric regions for biphasic responses and how they may be explored further.

Interestingly, in all models capable of biphasic responses studied above, almost all systems which are capable of exhibiting multistability are also capable of exhibiting it alongside biphasic response in the dose-response curve (see *Figure 2—figure supplement 1*).

## The network level

We now shift our perspective from the level of modification systems to the network level. From our studies above, we find that biphasic dose responses are readily seen in basic protein modification systems in cell signaling, requiring only minimal ingredients (enzyme sharing, sequestration, and multiple modification) which are widespread and readily encountered in cellular systems.

Cellular signal processing is largely dependent on networks involving protein modifications and thus the consequence of biphasic response at a node within a network can fundamentally alter network response. In addition to this, the widespread occurrence of biphasic responses (hormesis) at many levels is well-documented. These observations lead us to examine the consequences of such biphasic responses at the network level, by exploring the implications of biphasic interaction/regulation patterns. All in all, biphasic responses at the network level could be generated by (a) network topologies which facilitate this (b) complex biochemistry at a single node (as seen above), or (c) complex biochemistry (or implicit network structures) which may give rise to biphasic responses in a species not explicitly present as a node. It is this last case which we examine first, before turning to the other two cases. Here in effect, the problem reduces to the incorporation of the biphasic response in a network label/arrow (i.e. interaction pattern). For all relevant models, and how biphasic interaction/regulation is described, please see models and methods and *Supplementary file 1*, section 5.

### Organization of results

We probe the impact of a biphasic interaction pattern in a network. To do this, we first probe the effect of biphasic regulation by signal of a basic covalent modification cycle (a representative node of a network). We then explore the impact of biphasic regulation internal to networks, by examining ubiquitous network motifs (feedforward and feedback motifs). Finally, we explore the interplay of

biphasic regulation, interaction, and biphasic behavior intrinsic to a node in a network, the later arising from substrate modification biochemistry.

## Biphasic interactions within signal regulation

To explore this, we begin by analyzing how a simple signal can regulate a covalent modification cycle via a biphasic interaction pattern: (model N0, see *Figure 3A*). This simple interplay of a signal regulation and biphasic transduction can convert a saturated response into a biphasic response with a range of prolonged homeostatic responses in the concentration of the maximally modified substrate form, showing the potential for complicated signaling behaviors to emerge out of simple network considerations and biphasic interactions among nodes.

## Impact on network motifs

To explore this interplay further, we consider standard signaling motifs (feedback and feedforward) with biphasic interactions between the nodes (See models N3-N5 in *Figure 1—figure supplement 1*) and study how the biphasic interaction can alter expected network behavior. Networks motifs, such as feedforward and feedback networks, are recurring network structures in cellular signaling that are known to be responsible for key information processing characteristics such as multistability, biphasic responses, homeostasis, and oscillations.

The incoherent feedforward network motif (model N3), represents a simple network where a signal activates both a kinase and a phosphatase (and more generally activating opposing effects) and is known to present biphasic responses (*Varusai et al., 2015*; *Kim et al., 2008*). This nominal network behavior can be affected by the presence of a biphasic interaction between the nodes in three ways; (1) The behavior can be reinforced or strengthened leading to a more prominent overall biphasic response, (2) The two distinct biphasic responses (from the network structure and from the interaction between the nodes) can combine to give a multiphasic response, and (3) The two distinct biphasic responses can cancel out resulting in a simple monotonic dose response (see *Figure 3D*).

## Feedback motifs

The positive feedback network motif, characteristic of reinforcement, is known for introducing multistability (*Tyson et al., 2003*; *Ferrell et al., 2009*). Biphasic interaction between the nodes in such a network motif is capable of destroying such multistability arising from the feedback structure (see *Figure 3E*). Similarly the negative feedback network motif, characteristic of homeostatic response (*Nijhout et al., 2019*), can have this behavior impacted with the introduction of combinations of behaviors including homeostasis, multistability and monotonic responses (see *Figure 3F*). Particularly notable is the fact that multistability (usually associated with positive feedback) emerges from a negative feedback network in this case.

The basic intuition in both instances is that the biphasic interaction pattern causes deviation from expected behavior, simply because in a certain range, the nature of regulation changes from positive to negative. This negative regulation can combine with an overall negative feedback regulation pattern to give behavior reminiscent of a positive feedback (e.g. multistability).

The subversion of expected behavior and introduction of new behavior is seen even when the motifs involve open systems (see *Figure 3—figure supplement 2*). Analytical work reveals how negative feedback (involving an open system) can, in the presence of a biphasic interaction, present with multistationarity (which the motif is inherently incapable of in its absence; see *Supplementary file 1*, section 5).

## Impact on homeostatic regulation

We focus on a widely-studied motif incorporating homeostasis via integral control regulation which arises from a zeroth order reaction inherent in the system. Here, stable homeostasis is observed once the signal is above a certain threshold (the homeostatic state is unstable for lower signal values). We explore different aspects of biphasic regulation in interactions. If the signal regulation of the motif incorporates a biphasic interaction, this has the effect of creating both upper and lower limits of signal beyond which homeostasis does not occur. The presence of biphasic regulation within interactions intrinsic to the motif has two consequences (i) it limits the range of signal over which feasible

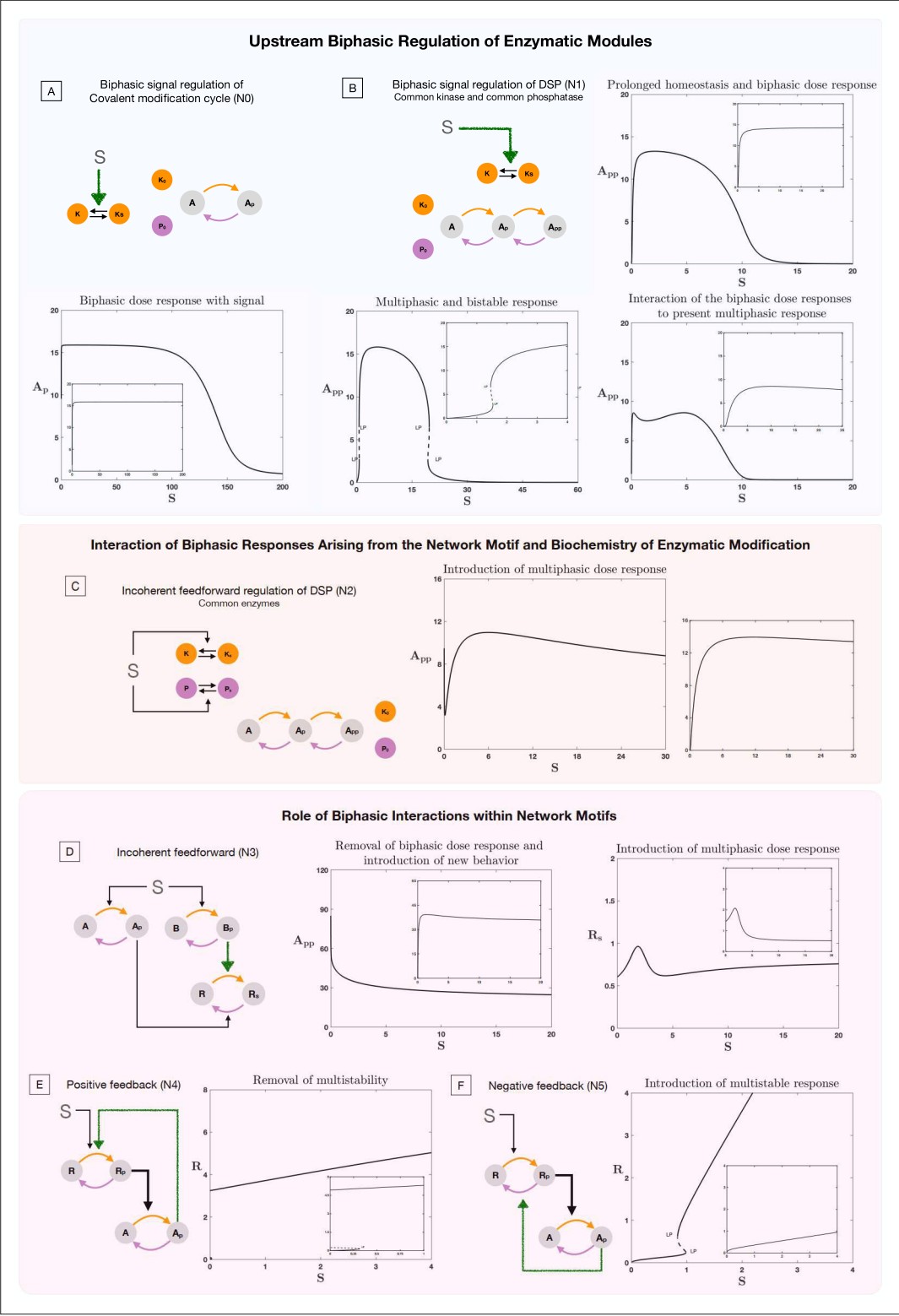

**Figure 3.** Role of biphasic interactions in network and motif response. The results presented in this plot show the consequence of biphasic signal regulation and biphasic response within interactions in enabling network response. In each plot the inset image represents the basal behavior in the absence of biphasic signal regulation or biphasic interaction, for the same kinetic parameters of the system. (**A–B**). Role of biphasic signal regulation in perturbing behavior of enzymatic systems, (**A**) Covalent modification cycle and (**B**) double site modification (DSP) (common enzymes). (**A**) shows how biphasic signal regulation can enable homeostatic (prolonged flat region) and biphasic

*Figure 3 continued*

response from a covalent modification cycle (note that the response is an increasing function of dose for small signal values). (**B**) shows how biphasic signal regulation can enable not just biphasic behavior (top right panel) but multiphasic behavior (bottom right panel), and result in a dose response curve with multiple multistable regions (indicated by pairs of limit points) along with a biphasic response (bottom left panel). (**C**) contrasts this with the response from incoherent feedforward regulation of the DSP (a driver of biphasic responses), which can similarly present multiphasic responses (note that the curve decreases before increasing and decreasing again) (plot to the right shows behavior without incoherent feedforward regulation, where such behavior is absent). (**D–F**) *Effect of biphasic interaction on common network regulatory motifs.* Biphasic interaction can alter the fundamental feature of feedforward motifs in multiple ways. (**D**) In incoherent feedforward networks, it can destroy the expected biphasic response (left panel) and also enable the creation of multiphasic responses (right panel). Similarly, in feedback networks, it can destroy multi-stability in a positive feedback motif (**E**). On the other hand, it can erode homeostatic responses and enable multistability in negative feedback motifs (the later shown in **F**). [Colored straight arrows in network schematics represent biphasic interactions. LP: saddle-node bifurcation, solid lines, and dotted lines denote stable and unstable steady state, respectively].

The online version of this article includes the following figure supplement(s) for figure 3:

**Figure supplement 1.** Effect of biphasic response in interactions within feedback network motifs (open systems): The figure shows the introduction of novel behavior (bistability in NFB, see (**B**)), and the removal of expected behavior (bistability in PFB, see (**A**)) even when the biphasic responses in interactions are present within open system feedback network motifs.

**Figure supplement 2.** Effect of biphasic response in interactions within integral feedback control motif: perturbation of expected homeostatic response.

---

homeostatic steady states occur (see *Figure 3—figure supplement 2*) (ii) it results in two steady states, one of which is shown to be necessarily unstable.

The above results indicate how hidden biphasic responses between nodes in networks can not only fundamentally alter expected network responses, but can also lead to incorrect inferences regarding the structure of the network (e.g. inferring networks from data), its information processing capability and the role of its constituent elements.

## Complex signal regulation and interaction of biphasic responses from distinct sources

Cellular signaling often involves both complex network regulation and additional complexity at the level of a node, with the node itself being capable of complex behavior (such as the DSP system). This implies the possibility (in addition to the cases considered above) of biphasic responses both in interactions as well as within a node in a network.

To explore this interplay further, we consider the signal regulation of DSP (with common kinase and phosphatase). This allows us to explore the interplay of biphasic response arising from two fundamentally different sources, (1) the network (for instance incoherent feedforward signal regulation) or biphasic response within an interaction between nodes, and (2) the intrinsic biochemistry of the node (DSP, is itself capable of exhibiting biphasic response - see *Figure 3B, C*).

We incorporate these aspects in simple models N1 and N2 (see *Figure 1—figure supplement 1*); In the former instance (model N1, *Figure 3B*) the biphasic response is inbuilt into the signal regulation activating kinase, where as in the latter (model N2, *Figure 3C*), the biphasic arises from the incoherent feedforward network structure (signal activating both enzymes).

When the biphasic interaction from the network arises directly from the regulating signal, the combination of biphasic responses from the network and the node can result in (i) introduction of biphasic responses (when the DSP does not present it inherently) (ii) strengthening the biphasic response and also generating distinct zones of multistability and homeostasis (*Figure 3B*), when the DSP presents biphasic responses independently. When the biphasic behavior at the network level arises out of the incoherent feedforward regulation (*Figure 3C*), similarly the behavior could be strengthened or destroyed. In both instances, multiphasic response is also possible.

Thus beyond simple signal regulation, minimal complexity at the level of the biochemistry of the node and biphasic responses (either within the signaling interaction, or arising from network structure) can lead to a plethora of unexpected and non-intuitive signaling behavior.

## An exemplar case

### Erk Regulation

The ErK pathway is an example of a pathway of central biological interest, which is well studied, contains the biochemical building blocks studied above and is known to exhibit biphasic responses. This makes it an excellent exemplar case worth investigating in detail. We explored the propensity of the established Erk pathway to present substrate biphasic responses (with total amounts of Erk) and enzyme biphasic responses (with total amounts of Mek) in the concentration of the maximally modified Erk (pYpTErk). Studies in the past have focussed on the capacity of the Erk regulatory pathway to show substrate biphasic responses computationally (*Witzel and Blüthgen, 2018*).

Our analysis reveals that the Erk pathway is capable of exhibiting both substrate and enzyme biphasic responses in the maximally modified substrate form (see *Figure 4A and B*). The system is in fact capable of biphasic dose response with both doses simultaneously, for the same underlying kinetic parameters (see *Figure 4C*). Analytical work further reveals that substrate biphasic responses are guaranteed to exist for some total amounts of Mek and phosphatase, irrespective of intrinsic kinetic parameter values. This is indicative of a degree of robustness in this behavior, as intrinsic kinetic parameters are varied, which is not apparent in a purely computational study. Analytical work revealed an explicit condition (dependent on total amount of phosphatase and kinetic parameters) that guarantees the existence of enzyme biphasic response. This condition is, however, only sufficient and not necessary for observing this (see *Supplementary file 1*, section 6) These results are summarised and consolidated in *Figure 4*. The consequences of these results for both systems and synthetic biology are discussed in the conclusions. We note here that the Erk system represents examples of biphasic interaction and regulation in cell signaling, both for its regular downstream pathways, as well as pathways it interacts with as a consequence of cross-talk.

It is worth pointing out that the Erk network has the structure of a random substrate double site modification system, with partial irreversibility. Having established the result above, along with analogous results for ordered modification systems (one leg of the Erk network), we can assert that essential insights are relevant even if one of the irreversible steps is made reversible.

All in all, we can conclude that the commonality of kinase in different modification steps and the commonality of phosphatase in demodification steps provides the (different) in-built competing effects for enzyme biphasic and substrate biphasic responses to be realized.

## Discussion

Biphasic dose responses are ubiquitous at all levels of biology, and in cell signaling and biochemical networks in particular. In this context, the overall attenuation with dose suggests beneficial safeguarding mechanisms which limit and set bounds on output and are associated with 'optimal' ranges of activation. In other instances, the departure from monotonic dose-responses may be undesirable (particularly in pharmacology and drug treatment) (*Calabrese, 2010*; *Randall et al., 1947*; *Shanmugam et al., 2022*). Given the widespread occurrence of such responses, we aimed to study this with a unified approach (both enzyme and substrate biphasic responses), at different levels (biochemical modification, network, specific exemplar systems) using analytical and computational approaches to unambiguously establish the presence or absence of these responses. This provides a unified synthesis of different types of biphasic responses in cell signaling and post-translational modification systems, their diverse origins, and consequences. Our focus was on how such responses could be obtained from the most basic aspects of intrinsic kinetics of substrate modifications and network organization, complementing studies of other sources of these responses (scaffolding, protein structure modification, product inhibition, competing pathways [*Levchenko et al., 2000*; *Karanicolas and Brooks, 2003*]).

Our analysis focused on a suite of commonly occurring substrate modification systems, using broadly employed models, and revealed explicitly which systems were capable of generating biphasic responses. This suite of substrate modification systems started with the basic covalent modification cycle, and considered basic extensions therefrom involving multiple modifications of the substrate (involving either common or distinct enzymes), enzymatic modification cascades, and coupled modification cycles. Since the sharing or enzymes and/or substrates is a key driver of biphasic responses,

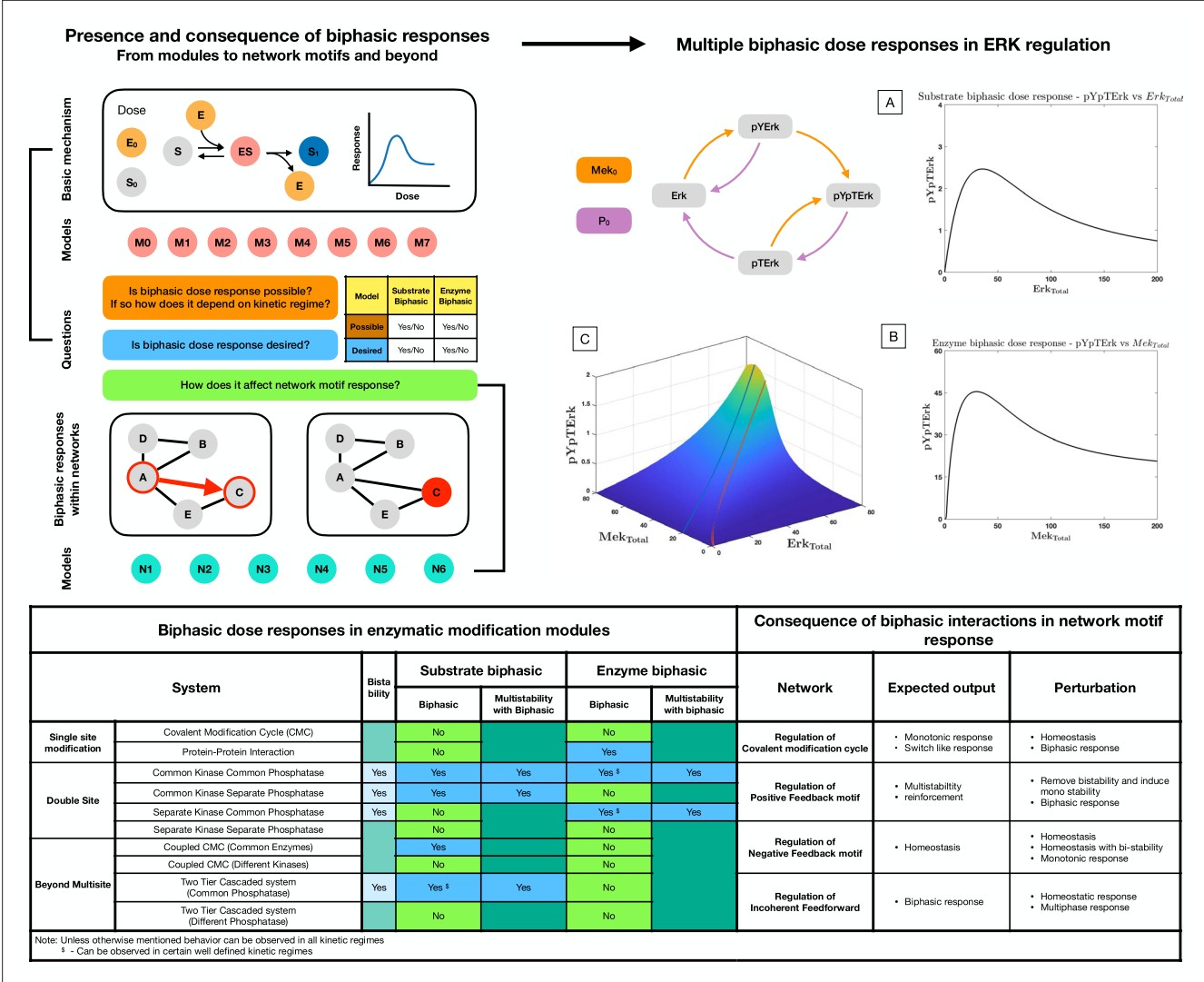

**Figure 4.** Presence and consequence of biphasic dose responses in cellular signaling arising from enzyme-mediated substrate modification: *Top left hand panel*: Schematic representation of the study. The basic mechanism of enzyme-mediated substrate modification (when present in different reversible modification cycles) allows for simple biochemical systems to present biphasic dose-response behavior. This is analyzed in a series of commonly encountered biochemical building blocks of cell signaling (M0-M8, see *Figure 1—figure supplement 1* for detailed schematic). In each case, we assess whether the system exhibits biphasic behavior with respect to variation of total amounts of enzyme and substrate as doses, and if it does so robustly. The table below provides a classification system to determine the different possibilities of biphasic responses (enzyme/substrate) vis-a-vis the desirability/undesirability of biphasic responses. This provides a framework (in synthetic biology) for determining what changes need to be made to a system, to obtain desirable characteristics. We then extend the study to consider the consequence of such biphasic dose responses in network motif structures (either within nodes, or in interactions–colored in red) and study how biphasic interactions can alter expected outcomes from these systems (N1–N5). *Top right hand panel*: Case study: Extracellular signal-regulated kinase (Erk) Regulation. The ERK regulatory network is capable of both exhibiting substrate and enzyme biphasic response in the double phosphorylated Erk (pYpTErk) in the steady state dose response with $Mek_{Total}$ (Enzyme biphasic) and $Erk_{Total}$ (Substrate biphasic). A-B shows an instance of enzyme and substrate biphasic response, respectively. (**C**) shows how the steady state concentration of pYpTErk changes with $Mek_{Total}$ and $Erk_{Total}$ (for a fixed kinetic parameter set). This indicates the presence of simultaneous substrate and enzyme biphasic response in the system (red and blue lines indicate the peak concentration of pYpTErk achieved for a given amount of total substrate and total enzyme amounts, respectively). *Bottom panel. Table*: Detailed summary of the various results discussed in the manuscript, including the possibility and impossibility of specific dose-response behaviors in a model, and the different ways in which the expected outcome of a network motif structure is undercut by biphasic responses within interactions. *Note*: These results discuss analytical results which can be found in *Supplementary file 1*.

this suite of systems provides a systematic basis for exploring how biphasic responses may emerge at the substrate modification level.

For each of these systems, a key aspect of our analysis was determining, to what extent substrate modification kinetic parameters precluded or enabled the possibility of biphasic responses. The parameters involved the intrinsic kinetic rate constants, as well as total amounts of enzyme(s) and substrate (apart from the dose). Our analysis was able to completely characterize the impact of intrinsic kinetic rate constants (the largest group of parameters and the hardest to vary experimentally). Generally, three possibilities were encountered: (i) Biphasic responses were precluded irrespective of intrinsic kinetic rate constants, pointing to a structural obstruction to obtaining biphasic responses (ii) biphasic responses were possible for any values of the intrinsic rate constants, and in each case, suitable values/ranges of total amounts of enzyme/substrate could be found to allow for biphasic responses. This corresponds to a widespread occurrence of biphasic responses in the parameter space of intrinsic kinetic rate constants (iii) biphasic responses could be found in specific (broad) regions of the parameter space of intrinsic kinetic rate constants, These regions are explicitly determined, and importantly, place no restriction on the absolute values of rate constants (only on relative values).

Building on the analysis above further parametric analysis was performed using analytical, semi-analytical, and computational means, We were able to assess the roles of species total amounts as well. In each case where biphasic responses were possible, We were able to show that they were obtainable in parameter regimes which were reasonable in both natural biology and synthetic biology.

By focussing on substrate modification systems which are building blocks for cell signaling and post-translational modification, we are able to establish clear boundaries and transitions between the non-occurrence, occurrence, and widespread occurrence of different biphasic responses arising from the intrinsic kinetics, and the structural features responsible for each of these possibilities. For these systems, some exhibited biphasic responses only to the enzyme, while others did so only to the variation of the substrate amount. Other systems exhibited biphasic responses to both substrate and enzyme, representing a robust multi-pronged in-built safeguarding mechanism. For some systems, we were able to demonstrate that biphasic responses are precluded, irrespective of parameters. The implication of such a result is that if biphasic responses were observed experimentally, it would have to be due to some other factor, either at the network level (eg. network regulation) or at the modification level (e.g. scaffolding).

## Drivers of biphasic responses in biochemical systems

Ultimately, biphasic responses arise from in-built competing effects. It is interesting to note that enzyme sharing (a simple consequence of the organization of cellular networks with different roles for both enzymes and substrates) provides these in-built competing effects resulting in biphasic responses, enabling them when they didn't exist otherwise. The sequestration of enzymes and substrates in complexes is a key ingredient here, along with enzyme sharing. In each case, we can trace the inbuilt competing effect. In some instances (covalent modification cycle with additional interaction, enzyme biphasic in double site modification system with shared phosphatase, substrate biphasic response in a two-tier enzymatic cascade with common phosphatase), it is possible to intuitively see this in-built competition (see Results). In other cases (e.g. substrate modification in double site systems with shared kinase, coupled covalent modification cycles), a competing effect with sufficient non-linearity (arising from some co-operative effects) is the basis for generating biphasic responses (see Results).

Multisite modification, especially with commonality of enzymes, promotes biphasic responses. Interestingly, a commonality of kinase promotes a substrate biphasic response, while a commonality of phosphatase promotes an enzyme biphasic response in multisite substrate modification systems. A commonality of both kinases and phosphatases thus enables both substrate and enzyme biphasic responses. The effect of these ingredients also depends on the underlying modification systems: for instance, a commonality of phosphatase promotes a substrate biphasic response in enzymatic cascades (in contrast to its creation of enzyme biphasic responses in multisite modification systems). Similarly, for random (i.e. non-ordered) modification systems, even distinct enzymes performing each modification can generate both biphasic responses. In many instances, the biphasic behaviour observed is surprisingly robust, and for any given kinetic parameters, a suitable choice of enzyme and substrate amounts (experimentally accessible factors) guarantees that it will occur.

## Biphasic responses amidst other information processing characteristics

Interestingly, the biochemical building blocks we have examined giving biphasic behavior can intrinsically also generate other complex behavior such as bistability (see table in *Figure 4*). In almost all cases (except the ordered double-site modification mediated by a common kinase/phosphatase pair), we have found the simultaneous co-existence of both behavior possible in a dose-response. In the context of ordered double-site substrate modification (mediated by a common kinase/phosphatase pair), we uncover an unusual instance of a system where one of a suite of non-trivial systems behaviors (in this case biphasic responses and bistability) is guaranteed across the intrinsic kinetic parameter space, arising from the largely complementary parametric requirements for each of these behaviours.

## Biphasic regulation and interaction in networks

We then investigated the impact of such biphasic responses in networks by associating certain interactions (which may contain multiple species or subnetworks implicitly) with biphasic responses. We find that biphasic interaction patterns can completely abolish expected behavior in networks, and generate unexpected new behavior which may not be characteristic of the network (e.g. bistability in negative feedback networks). It can also significantly impact the expected functioning of networks in different ways (exemplified by its impact on homeostatic control networks). We also explored the interplay of two sources of biphasic behavior: the network regulation and substrate modification and show that this could lead to reinforcement, cancellation, or even multiphasic or other (e.g. multistable) responses. Consequently, the presence of biphasic responses has an even greater import on the behavior of signaling networks than may be expected from basic considerations.

## Methodology of analysis and testable predictions

Our analysis makes a number of predictions regarding the absence, presence, and robust occurrence of biphasic responses in a variety of biochemical modification systems, as well as consequences of biphasic regulation in networks. This results in a number of experimentally testable predictions, for example, the realization of enzyme or substrate biphasic responses in different systems (including Erk signaling) which can be obtained by suitably varying enzyme and/or substrate amounts, or the generation of unexpected network behavior as a consequence of biphasic regulation. These could be tested directly in the relevant system (e.g. Erk or other basic modification system) using experiments both in cells and in test tube settings. The consequences of biphasic regulation in networks could be tested using networks containing biphasic regulation, which could be engineered (and tuned) synthetically. Our analysis also makes a number of conclusions regarding the possibility of different types of biphasic responses in different building blocks of substrate modification. In each of these instances, the relevant building block can either be isolated from concrete cellular pathways, or built synthetically. Our analysis already predicts which types of systems are capable of biphasic responses (to variation of enzyme or substrate). For the systems where intrinsic kinetic parameters play no role in obstructing the possibility of biphasic responses, our analysis and its extensions (semi-analytical and numerical) can predict that a biphasic response can be obtained and also how substrate/enzyme total amounts must be varied to obtain a biphasic response: this later prediction requires a knowledge of the intrinsic kinetic rate constants. For cases where the possibility of biphasic responses occurs in a subset of the parameter space (of intrinsic rate constants), our analysis predicts whether or not a biphasic response is possible (this requires a knowledge of intrinsic rate constants) and for cases where it is possible, can be used to determine total amounts of substrate or enzyme where that is possible. Furthermore, we can also make predictions which are independent of parameters. Consider cases where there is a parametric restriction in obtaining biphasic responses: say enzyme (kinase) biphasic responses in the DSP system with common kinase and common phosphatase. We can predict that if an enzyme biphasic is obtained (i.e. for the concentration of the fully modified form as total amount of kinase is varied), then a biphasic response in the concentration of the unmodified form as phosphatase total amount is varied is also possible (for the same set of intrinsic kinetic parameters). This is because a parametric condition which allows for a biphasic response in one direction (forward) say, is exactly what allows it in the opposite direction. This emerges from analyzing the relevant parametric expression (analysis of an enzyme biphasic in the reverse direction, amounts to analyzing the same system, with altered labels and parameters).

Our methods of analysis, combining both analytical and computational analysis could also be deployed for deconstructing and analyzing the possible occurrence and robustness of different biphasic responses in other systems comprising network regulation and substrate modification systems (e.g. p53, Ras, MAPK pathways).

All our analysis of systems has focussed on steady states. The implicit assumption here is that for various processes of interest, the systems under consideration quickly approach the steady state. Therefore, even though other processes of interest may occur transiently, a steady state analysis of these sub-systems provides useful insight.

We now discuss the relevance of our results for both systems and synthetic biology.

## Systems biology

### Basic cellular biochemical building blocks feature in-built safeguards against protein overactivation and overexpression

The first basic observation is that many ubiquitously occurring biochemical building block structures present either enzyme biphasic or substrate biphasic behaviour, and in some cases both behaviour, and this arises from the presence of key basic ingredients. An example is enzyme sharing which is common in cellular networks, especially since kinases significantly outnumber phosphatases, leading to phosphatases deployed in different kinase mediated reactions (*Ghaemmaghami et al., 2003*). Interestingly while certain building blocks do not admit biphasic responses (unless some additional feature such as substrate inhibition occurs), others do, and in certain cases, they can even be obtained just by tuning substrate and enzyme total amounts (completely independent of intrinsic kinetic parameters). Since these building blocks are ubiquitous in cells, this already indicates a number of in-built sources of safeguarding against overexpression of active entities in signaling networks, beyond network regulatory structures.

As an illustration, we have investigated a single tier of the Erk signaling network and demonstrated how this exhibits robust substrate biphasic responses (without restriction on intrinsic kinetic parameters, in contradiction to the picture suggested in *Witzel and Blüthgen, 2018*) and can also exhibit enzyme biphasic responses in broad parameter ranges. This illustrates how a single level of a specific signaling pathway may incorporate multiple levels of safeguarding and that the structure of the biochemical reactions involved provides cells (and evolution) with easily tuneable factors to realize this safeguarding. This also provides an example of biphasic regulation/interaction in networks, by considering the various downstream targets of Erk (as well as elements impacted by Erk cross-talk). Another consequence of the above results is the possibility of biphasic dose-responses occurring, which may be not (particularly) desirable, but are present due to the characteristics of the basic building blocks. This could, in some contexts, illuminate why such responses may occur.

### Inferring drivers of biphasic response in networks

The fact that there are multiple intrinsic sources of safeguarding and biphasic responses in signaling networks (i.e. even intrinsic to a node in these networks), suggests that there may be a hierarchy of such mechanisms–ranging from network structures (such as incoherent feedforward networks) to additional ingredients in modification systems (such as scaffolding) to the intrinsic modification systems themselves. This is especially relevant in inferring network structures from data (for example, observed biphasic responses) and suggests that the inference process must be able to account for different sources of such behavior. Our analysis suggests that focussed experiments can discriminate between these different sources of biphasic responses and simple controlled test tube experiments can reveal the presence of intrinsic biphasic responses. Such experiments would be very valuable as part of the inference process.

### Consequences and implications for network behavior

Analyzing the consequences of biphasic responses at the network level yields further insights. We have already demonstrated that the presence of biphasic responses of the kind studied leads to biphasic interactions between network nodes and this can significantly alter network behavior, even resulting in behavior which might be expected from a very different network structure. Biphasic responses can be used in networks to access specific downstream behavior in specific ranges of inputs. As such it can combine with complex dynamical behaviour (e.g. bistable switching, oscillations) in different

ways giving rise to behavior which may appear unintuitive, if the presence of biphasic behavior is not recognized.

The presence of safeguarding mechanisms/biphasic behavior at different locations in a network has important consequences for the overall robustness of the network as well as the maintenance of behavior within bounds. Our results revealed how biphasic responses at different levels (e.g. network and substrate modification) may either reinforce or cancel one another, suggests that the implications of different sources and locations of biphasic responses/safeguarding in a network, need to be carefully studied via a dedicated systems analysis. The analysis of how distributed sources of such safeguarding (both enzyme and substrate biphasic behavior) determines the overall safeguarding behavior in the network and keeps relevant species behavior within suitable bounds is a topic of future investigation.

### Biphasic responses: Matching capacity with desired functionality

Natural and engineered biology and the emerging discipline bridging the two, motivates the need to examine the presence or absence of biphasic responses relative to desired functionality. To start with, if we focus on a given kind of biphasic response (either enzyme biphasic or substrate biphasic) then we have four distinct possible cases depending on the presence or absence of this behavior and whether its presence is desired or not (Table in upper left panel, *Figure 4*). Two of these cases (biphasic response present, desired) (biphasic response absent, undesired) represent desirable outcomes, and the key insight in the former case is that this may be obtained in broad swathes of parameter space by just varying total amounts of enzyme and substrate. The other two cases (biphasic response absent, desired), (biphasic response present, undesired) represent a mismatch between desired functionality and capability. In the former case, (apart from possible tuning of enzyme or substrate amounts) alternative structures can be engineered or realized in evolution, for instance, the creation of incoherent feedforward networks (for realizing enzyme biphasic responses) or the incorporation of extra structures in the substrate modification systems (for realizing substrate biphasic responses). In the latter case, a tuning of total enzyme and substrate amount can be used to prevent biphasic responses in desired ranges of operation, and if this can be done consistent with information processing goals, a desired outcome can be obtained.

The same picture can now be expanded to consider the possibility of both enzyme and substrate biphasic responses in relation to whether those biphasic responses are desirable or not (see *Figure 4*). This results in 16 cases, four of which correspond to both desired outcomes being met, four of them corresponding to both desired outcomes not met, and eight cases where one desired outcome is met and one is not. In order to 'engineer' the system towards desired goals, different approaches can be taken depending on the specific case and mismatch. Clearly depending on the context, the managing of desired outcomes requires a combination of structural and (easy to manipulate) parameter manipulation and this may involve a careful optimization to keep it consistent with the information processing goals. Our analysis provides a basis from which to evaluate what steps may have been incorporated in evolution in different contexts to achieve safeguarding against over-expression or overactivation, including the extent to which intrinsic capabilities for such responses have been used. This needs to be evaluated on a case by case basis.

## Synthetic biology

### Structural and parametric analysis of a variety of systems transparently reveals engineering design principles

The results have further specific consequences for synthetic biology. Biphasic responses have been generated from synthetic gene regulatory incoherent feedforward circuits (*Basu et al., 2004*) and synthetic microbial communities (*Zong et al., 2022*). There have been multiple recent efforts to engineer biochemical circuits to obtain complex information-processing behavior (*Maguire and Huck, 2019*; *Helwig et al., 2018*; *Holmqvist et al., 2012*) and these include engineering substrate modification as well (*Valk et al., 2014*; *O'Shaughnessy et al., 2011*). Our demonstration of basic building blocks generating biphasic responses is of particular use in engineering biochemical networks. Of particular relevance is the presence and guarantee of obtaining such behavior, and in some cases the widespread occurrence of such behavior, which can be accessed by varying simple easy to vary factors (substrate and enzyme amounts). Our analysis in totality across systems provides structural

requirements for the generation of such behaviour and where necessary also provides kinetic parameter ranges where this may be obtained. This combination of structural and parametric criteria provides engineering design principles for creating such behaviour from basic biochemical building blocks. Furthermore, our analysis can be used to engineer both substrate and enzyme biphasic responses, representing a robust safeguarding mechanism for preventing overexpression of specific substrate forms. Interestingly, in addition to our analysis of building block circuits, our analysis of a key layer of Erk signaling reveals such robust safeguarding mechanisms, making this an appealing candidate for incorporating into synthetic circuits.

## A comparative analysis of different systems reveals benefits and tradeoffs for design

Our examination of different building block structures in comparison with one another provides further insights. For instance, the presence of a 'dead-end' complex (protein-protein interaction system) reveals a robust presence of enzyme biphasic responses, while double site substrate modification with either common kinase or different kinases but common phosphatases provides enzyme biphasic responses in a substantial region of kinetic parameter space (the region in fact being exactly the same). A comparison of the latter two cases suggests that the separate kinase common phosphatase system provides additional tuneability, along with access to other behavior, with no loss of capability in terms of biphasic responses. Interestingly, a number of substrate modification systems: double site modification with common kinase common phosphatase, common kinase separate phosphatase, as well as coupled modification cycles sharing a common kinase and phosphatase all yield robust substrate biphasic responses across intrinsic kinetic parameter space. Comparing the first two cases, the possibility of having different phosphatases may again provide additional tuneability. In the third case, one of the cycles is guaranteed to give substrate biphasic responses, the identity of which is determined by an explicit analytic expression involving kinetic constants. Taken together, this illustrates how our analytical work across different systems reveals tangible structural and organizational insights directly relevant to design.

## Engineering network behavior

The analysis here can also be used to shape overall network/circuit behavior. Modules exhibiting biphasic responses with tuneable characteristics can be designed and incorporated into chemical reaction networks (either naturally occurring or itself synthetic) to shape outcomes, regulate behavior, and create different safeguarding mechanisms against over-expression in networks. Interestingly some preliminary analysis (*Ramesh et al., 2023*) shows how basic modules studied here when combined with positive and negative feedback can in fact maintain, not just the biphasic response, but specific characteristics such as the peak amplitude, thus testifying to both the robustness of the insights and its value in engineering design.

All in all, our unified exploration of enzyme and substrate biphasic responses through the analysis of ubiquitous building blocks in networks opens up new vistas for exploration in natural biology, engineered biology, and the emerging area straddling the two.

# Materials and methods
## Systems considered

The systems considered and investigated in this paper span a range from basic biochemical building blocks to basic network motifs. With regard to basic biochemical building blocks, we examine a suite of basic biochemical systems involving enzyme-mediated protein modification, all of which build on basic covalent modification cycles. Thus we consider covalent modification cycles (both basic cycles and those with 'dead-end' complexes), double site modification systems (ordered and distributive) where different (de)modifications are effected by either the same or different enzymes. We also consider two-step enzymatic modification cascades where the phosphatases in the two steps may be either the same or different. We also examine coupled modifictaion cycles involving shared enzymes. Taken together, these systems (models M0-M7) represent a suite of basic biochemical building blocks (incorporating different kinds of modification 'logic') and their extensions, which are repeatedly encountered in biochemical pathways. It is worth pointing out that by considering these various

building blocks (themselves, amongst the simplest extensions of covalent modification cycles), it is possible to obtain transparent insights (arising from the most basic considerations of the kinetics) into key structural features enabling different kinds of biphasic responses (see *Figure 2*).

Building on this, we also consider simple signaling network motifs (feedforward and feedback networks). Here, the goal is to explore how biphasic regulation which may be present within the network (due to regulatory steps implicit in the network) may impact network behavior. Thus, we examine these motifs with biphasic interaction between nodes of the network motif (models N0, N3-5). Finally, we also examine cases which bring elements of network regulation along with the biochemical protein modification systems studied above, together. To do this we focus on the case of double site protein modification system, and examine both biphasic regulation of the kinase, as well as incoherent feedforward regulation (via incoherent feedforward networks). These are examined in models N1 and N2.

Complementing the above systems, we study a biochemical exemplar system, the Erk regulatory pathway (model adopted from *Witzel and Blüthgen, 2018*; *Rubinstein et al., 2016*). A summary schematic of the various models used is presented in *Figure 1—figure supplement 1*.

## Mathematical modeling

We first describe the modeling of the basic biochemical building block systems considered above. The systems considered above are modeled as a system of ordinary differential equations using simple mass action kinetic description for the elementary reactions. Each (de)modification step involves three elementary reactions, the reversible binding (and unbinding) of the free active enzyme with the substrate to form an enzyme substrate complex, followed by the irreversible catalytic reaction to release the free active enzyme and the (de)modified substrate (see *Figure 1*). An abundance of ATP is implicitly assumed. Using such a description for every (de)modification step, the mathematical models for the biochemical systems are constructed. Since the total amount of substrate(s) and enzymes are also conserved in the system, the ODEs are also associated with the respective conservation equations. We note that the models of the biochemical systems generated, represent widely employed models of such systems. We do not incorporate other effects such as product inhibition or scaffolding.

We present below how the covalent modification cycle (where a substrate gets modified and demodified by a kinase and a phosphatase, respectively; model M0) can be described mathematically using such a description.

$$\frac{d[A]}{dt} = -k_{b1}[A][K] + k_{ub1}[AK] + k_2[A_pP]$$

$$\frac{d[AK]}{dt} = k_{b1}[AK] - (k_{ub1} + k_1)[AK]$$

$$\frac{d[A_p]}{dt} = k_1[AK] - k_{b2}[A_p][P] + k_{ub2}[A_pP]$$

$$\frac{d[A_pP]}{dt} = k_{b2}[A_p][P] - (k_{ub2} + k_2)[A_pP]$$

$$\frac{d[K]}{dt} = -k_{b1}[A][K] + (k_{ub1} + k_1)[AK]$$

$$\frac{d[P]}{dt} = -k_{b2}[A_p][P] + (k_{ub2} + k_2)[A_pP]$$

$$A_{Total} = [A] + [A_p] + [AK] + [A_pP]$$

$$K_{Total} = [K] + [AK]$$

$$P_{Total} = [P] + [A_pP]$$

The mathematical models for the other systems considered are constructed in a similar manner and simply involve a combination of such descriptions for individual modification/de-modification steps In using simple mass action kinetics to represent the kinetics of the individual elementary steps (which are part of an overall modification step), we make no a priori assumption regarding the kinetic regime of the enzyme action, and thus the results are broadly representative of the overall behavior of the system. The computational models created in MATLAB are cross-validated with model descriptions generated in COPASI (*Hoops et al., 2006*) (which only requires definition of the reaction schematic for model generation).

### Modeling biphasic interactions between nodes in a network

The biphasic interactions between nodes $A \rightarrow B$ in a network was implemented by simply incorporating a biphasic function connecting the activation of B with the concentration of A. For this purpose,

we use a Poisson function, given by the following mathematical expression $k_1[A]e^{-k_2[A]}$. This is in contrast to a basic (monophasic regulation) for instance by a linear function ($k_2[A]$). We note that the particular function we use to describe the biphasic regulation is representative, and the essential results and insights do not depend on the particular choice of biphasic function.

When choosing parameters in the context of systems with biphasic response in the interaction ($k_1$ and $k_2$), in order to maintain parity and a like-for-like comparison between the system with the biphasic regulation and without it, we ensure that the average strengths of the interaction terms remains the same when averaged over the range of total signal activation. Further details on how this is implemented is provided in the *Supplementary file 1*, section 5.

## Parameters

The models constructed have a number of parameters, involving both kinetic constants and total amounts of substrate(s) and enzymes. It is useful to draw a distinction between these two classes of parameters for multiple reasons. (1) The total species amounts represent easily accessible levers that can be tuned, while the kinetics are intrinsic to the system and generally less accessible. (2) In the context of biphasic dose responses, the total amounts represent doses to the system and thus are natural experimental factors that change, and consequently are basic parameters with respect to which bifurcation analysis is carried out. (3) In many systems (e.g. double site phosphorylation) the possibility of different behavior (e.g. bistability) is actually determined by constraints on intrinsic kinetic parameters, and once these constraints are satisfied, it is possible to vary substrate and enzyme amounts suitable to ensure that the desired behavior is realized. Thus, there is a hierarchy in the set of parameters to enable different information processing behavior.

## Doses

Enzyme-mediated protein modifications have two natural doses (with respect to the maximally modified substrate form), i.e., (1) Total substrate amount, and (2) total enzyme amount.

## Approach

The approach to assessing biphasic response in the networks is as follows.

We begin by analytically characterizing the presence or absence of the biphasic behavior for the relevant substrate concentration with the dose, and in the process obtain binary answers to the feasibility of observing the behavior in parameter space for a given model. In the instance where the behavior is shown to be present, we show evidence of it computationally with a 1-parameter bifurcation analysis with the relevant dose. We further supplement this with analytical work to establish sufficient and necessary conditions (involving parameters, such as total amounts and kinetic constants) that guarantee the presence of biphasic dose response behavior. When the behaviour is not possible we analytically establish that this is the case irrespective of parameters.

### Bifurcation analysis

One-parameter bifurcation analysis was carried out computationally using the MatCont package in Matlab (*Govaerts et al., 2017*). In all cases showing the presence of biphasic response, the result is accompanied with analytical work establishing the presence of the behavior in broad regions of the parameter space (see below).

### Analytical approach to assessing biphasic response

Analytical work exploring the presence or absence of biphasic dose response (for a given choice of kinetic parameters and total amounts) relies on the mathematical requirement for a steady state of the system to satisfy the following expression at some finite concentration of dose $\frac{d[Var]}{d[Dose]} = 0$, should a biphasic response exist. Here, Var denotes the particular variable under consideration. Analytical work proceeds by exploring the possibility of the model satisfying this condition for some feasible steady state of the system. This is done by solving the system of ODEs of the model at steady state and ascertaining an expression for the above expression ($\frac{d[Var]}{d[Dose]}$) as a function of the steady state concentration of fewer variables and the parameters. This reduction allows us to ascertain whether

the behavior is possible/impossible across parameter space, and where possible ascertain conditions guaranteeing (and conversely precluding) the behavior. The entirety of this analysis is carried out in the computational platform Maple (*Maplesoft, 2022*), and is presented in *Supplementary file 1*. This file is also supplied as a PDF for easy accessibility. A concise summary of the way analytical work is undertaken is discussed further below (see Summary of analytical approach).

Note: Our analysis throughout focuses on the existence of a biphasic dose response in the steady state of the system of the dose. We, however, make no comment or perform further analysis on the stability of the steady state when such a biphasic response is predicted. This, however, can be easily verified with computations and we supplement each of our analytical results with computational proof of the existence of the behavior for a range of parameters.

## Additional approaches

In the case of the biochemical substrate modification systems, our analytical work establishes to what extent intrinsic rate constants prevent or enable biphasic responses. We find three possibilities (a) biphasic responses are completely precluded (b) biphasic responses are possible for any values of the intrinsic rate constants: in this case, we can guarantee that there exist suitable values of total amounts of substrate and/or enzyme for this to happen (c) there is a region in the parameter space (of intrinsic rate constants), explicitly delineated, where biphasic responses are possible. Here again, we can guarantee that there are suitable values of total amounts of species to make this possible. In order to further investigate ranges of total amounts of species where biphasic responses are possible, we can start with the necessary requirement of a biphasic response (as in our analytical approach). This can be used further semi-analytically to determine total amounts of species where a biphasic response is realized. In particular, we can input kinetic parameters to determine the location of the biphasic peak (*Supplementary file 2*). As part of this, we also show that biphasic responses can be obtained in ranges of kinetic parameters seen experimentally. Here, we choose intrinsic kinetic parameters similar to those found in experimentally well-characterized reference systems (see *Witzel and Blüthgen, 2018*) and fix some enzyme amounts at physiologically reasonable values. We then obtain curves in the space of total amounts (see *Figure 2—figure supplement 3* and *Figure 2—figure supplement 4*) which correspond to the occurrence of a biphasic (peak) response. This can be confirmed by a computational bifurcation analysis.

## Summary of analytical approach

In this subsection, we provide a birds-eye view of the analytical approaches used in the various models. At the outset, we reiterate that our goal is to explore the presence or absence of both substrate and enzyme biphasic responses. In all the models, the focus is on a key substrate variable, as discussed in the Results. In all cases, the models involve solving for the steady states of the ODEs with appropriate conservation conditions. Typically each distinct type of enzyme and each set of substrates is associated with their conservation condition.

We will outline the approach in the case where there is a single kinase enzyme and a single set of substrates (examples of this include the simple covalent modification cycle, the protein-protein interaction model, the double site modification system with common kinase and common phosphatase and the double site modification system with common kinase and separate phosphatases). This is because the nature of the analysis is most transparent here. We then discuss how other cases involving more kinases and/or sets of substrates are studied.

The analysis progresses by a sequential elimination of variables at steady state starting with the concentrations of the various complexes in terms of concentrations of the constituent enzyme and substrate, and then writing other substrate concentrations in terms of the concentration of the substrate of interest. Then after exploiting any phosphatase conservation conditions, the steady state of the system can be determined from the two remaining conservation conditions $A_{con} = 0$ (substrate conservation condition) and $K_{con} = 0$ (kinase conservation condition). Note that the way the conservation condition is written, the total amounts of substrate $A_{tot}$ and kinase $K_{tot}$ are present in these expressions, linearly with coefficient –1.

If we denote the substrate variable by $A_{var}$ and the kinase variable by $K_{var}$, then the equations take the form $A_{con}(A_{var}, K_{var}, A_{tot}) = 0$ and $K_{con}(A_{var}, K_{var}, K_{tot}) = 0$. Now, suppose an enzyme biphasic

exists, then we require that $dA_{var}/dK_{tot} = 0$ at the biphasic peak. Differentiating both the conservation equations with respect to $K_{tot}$ results in two equations $\frac{\partial A_{con}}{\partial A_{var}}\frac{\partial A_{var}}{\partial K_{tot}} + \frac{\partial A_{con}}{\partial K_{var}}\frac{\partial K_{var}}{\partial K_{tot}} = 0$ and $\frac{\partial K_{con}}{\partial A_{var}}\frac{\partial A_{var}}{\partial K_{tot}} + \frac{\partial K_{con}}{\partial K_{var}}\frac{\partial K_{var}}{\partial K_{tot}} = 1$. The factor 1 on the RHS of the second equation is due to the presence of the factor $K_{tot}$ in the kinase conservation condition, linearly with coefficient –1.

Our analysis is based on examining whether such a pair of equations can be satisfied, with the requirement that $dA_{var}/dK_{tot} = 0$. The analysis, carried out in the Maple files in detail, proceeds as follows. We establish that the terms multiplying $dA_{var}/dK_{tot}$ in these two equations are necessarily finite (since these are rational expressions where the denominator is shown to be non-zero), and so when $dA_{var}/dK_{tot} = 0$, the associated terms can be eliminated. We are then left with two equations $\frac{\partial A_{con}}{\partial K_{var}}\frac{\partial K_{var}}{\partial K_{tot}} = 0$ and $\frac{\partial K_{con}}{\partial K_{var}}\frac{\partial K_{var}}{\partial K_{tot}} = 1$. Now there are different possibilities. 1. $\frac{\partial A_{con}}{\partial K_{var}}$ is shown to be necessarily non-zero (via a symbolic computation). This means that $\frac{\partial K_{var}}{\partial K_{tot}} = 0$. This, however, contradicts the second equation, since it can be shown that $\frac{\partial K_{con}}{\partial K_{var}}$ is finite. Thus the assumption of a biphasic response and biphasic peak is incorrect. This establishes that enzyme biphasic responses in this instance are ruled out. 2. $\frac{\partial A_{con}}{\partial K_{var}}$ can be zero. In this case, the possibility of the above contradiction is averted. There are different possibilities here (note that in all such cases, the problem reduces to the possibility of a polynomial expression in two variables being zero). (a) Structurally the polynomial expression contains two sets of terms with different signs. In this case, we establish that one can find suitable values for $A_{var}$ and $K_{var}$ so that a solution exists. There are ways to directly establish that a suitable solution exists for some values of these two variables. In such a case, we then use the conservation conditions with these values of $A_{var}$ and $K_{var}$ to establish that one can find suitable total amounts of substrate and enzyme $A_{tot}$ and $K_{tot}$ to ensure this happens. The conclusion is that for the given intrinsic kinetic parameters, one can always find total amounts of species (Substrate, enzyme) to ensure that a biphasic response is possible (b) Structurally, the polynomial expression does not necessarily separate into two sets of terms with different signs. However, it may do so, for certain ranges of kinetic parameters. In such a case, we establish that the equation can separate into two sets of terms with different signs for some regimes of intrinsic kinetic parameters. This then can be used to deduce the fact that a feasible solution of this equation exists in this regime of kinetic parameters. The upshot of this is that a biphasic response can be observed in certain ranges of intrinsic kinetic parameters, again by a suitable choice of total enzyme and substrate amounts.

## Comments

The same approach is used in all relevant cases of analysis. In the discussion below, the derivatives refer to derivatives of the substrate or kinase variables to the relevant substrate or kinase total amount. Note that the above analysis results in a set of linear equations for those derivatives, one homogeneous and one inhomogeneous.

1. First, in the above case, the possibility/impossibility of a substrate biphasic response is established in an exactly analogous way.
2. In the cases where we show a biphasic is possible, in many instances we are able to find values of the variables $A_{var}$ and $K_{var}$ where the associated equation has a solution (averting a contradiction between the two equations). This then establishes total amounts of substrate and enzyme where a biphasic peak can be expected. Numerical bifurcation analysis then establishes the actual presence of the relevant biphasic response.
3. In the above equation, the variable $A_{var}$ is the concentration of the substrate variable of interest. while the variable $K_{var}$ could either be the free kinase concentration, or in some instances, the ratio of free kinase to free phosphatase concentrations.
4. As the number of either substrates or kinases (or both) increases, the number of equations increases. However, they have the same structure. The LHS of the equations involves a linear combination of different derivatives of the relevant variables with respect to substrate or kinase amounts. The RHS of all but one of the equations is zero, and the RHS of one equation is 1 (when the derivative with respect to the total amount associated with that conservation equation is the focal point of interest).
5. In this case the impossibility of biphasic responses has been demonstrated in one of two ways. In both cases, we start with the assumption of a biphasic response, implying the possibility of a zero derivative of the relevant substrate variable to either a relevant kinase or substrate total

amount. (a) We progressively eliminate some of the derivatives in terms of others, until we obtain a set of equations, all but one, homogeneous. By assessing the coefficients of the derivatives in those equations, we show that a particular derivative (or set of derivatives) needs to be zero, which then makes the inhomogeneous equation impossible to satisfy (this is essentially an analog of the simpler case considered above) (b) In some cases, a consideration of two homogeneous equations themselves shows that they are impossible to satisfy simultaneously. In each case, this contradicts the assumption of a biphasic response

6. Likewise, the possibility of a biphasic response is established by showing that the simultaneous equations can have a solution. The essential insight is to show that the homogeneous equations can be satisfied in a way (by certain coefficients being zero), so that the inhomogeneous equation can also be satisfied. This then provides a condition to show that the biphasic response may either be observed (a) irrespective of intrinsic kinetic parameter values (relying only on choices of total amounts of species) or (b) requiring non-trivial restrictions on the intrinsic kinetic parameter values.

7. Note that all our analysis involves analyzing the first derivative requirement (to be zero). We do not explicitly check the second derivative condition (to show that there is a genuine change in sign of slope, as opposed to there being inflection point), simply because this is what can be expected generically (with the possible exception of 'rare' points in parameter space, which would correspond to sets of zero volume/measure). In this connection, we mention that when we make the statement that biphasic responses are observed everywhere in intrinsic kinetic parameter space, we mean that the responses are obtained essentially everywhere, with the possible exception of these special points. This does not affect our essential insight about robustness, Finally, our analysis does not explicitly assess stability. For all the cases analyzed, a predicted biphasic response has been shown computationally to be associated with a stable steady state over the range of parameters studied.

## Parameter values

*Figure 2*: Presence of substrate and enzyme biphasic dose responses in the commonly observed enzymatic cellular signaling motifs.

 A *Enzyme biphasic dose response in PPI model*

 $k_1 = 1$; $k_4 = 1$; $k_{b1} = 1$; $k_{b2} = 20$; $k_{b4} = 1$; $k_{ub1} = 1$; $k_{ub2} = 1$; $k_{ub4} = 1$; $A_{Total} = 30$; $P_{Total} = 1$;

 B *Enzyme and Substrate biphasic in DSP with common enzyme action*

 $k_1 = 0.1$; $k_2 = 1$; $k_3 = 1$; $k_4 = 1$; $k_{b1} = 1$; $k_{b2} = 1$; $k_{b3} = 1$; $k_{b4} = 1$; $k_{ub1} = 1$; $k_{ub2} = 1$; $k_{ub3} = 1$; $k_{ub4} = 1$; $K_{Total} = 2.5$; $P_{Total} = 1$;

 C *Substrate biphasic in DSP with common kinase and separate phosphatase action*

 $k_1 = 3.5$; $k_2 = 50$; $k_3 = 100$; $k_4 = 100$; $k_{b1} = 20$; $k_{b2} = 75$; $k_{b3} = 50$; $k_{b4} = 30$; $k_{ub1} = 1$; $k_{ub2} = 1$; $k_{ub3} = 1$; $k_{ub4} = 1$; $K_{Total} = 6$; $P1_{Total} = 1$; $P2_{Total} = 1$;

 D *Enzyme biphasic in DSP with separate kinase and common phosphatase action*

 $k_1 = 200$; $k_2 = 40$; $k_3 = 200$; $k_4 = 5$; $k_{b1} = 250$; $k_{b2} = 100$; $k_{b3} = 100$; $k_{b4} = 75$; $k_{ub1} = 1$; $k_{ub2} = 1$; $k_{ub3} = 1$; $k_{ub4} = 1$; $A_{Total} = 50$; $K1_{Total} = 18$; $P_{Total} = 30$;

 E *Substrate biphasic in the cascaded enzymatic model with common phosphatase*

 $k_1 = 80$; $k_2 = 35$; $k_{b1} = 200$; $k_{b2} = 200$; $k_{ub1} = 1$; $k_{ub2} = 1$; $p_1 = 5$; $p_3 = 60$; $p_{b1} = 40$; $p_{b2} = 25$; $p_{ub1} = 1$; $p_{ub2} = 1$; $A_{Total} = 45$; $K_{Total} = 10$; $P_{Total} = 16$;

 F Substrate biphasic in coupled covalent modification cycles with common enzymes

 $k_1 = 4$; $k_2 = 0.1$; $k_{b1} = 0.9$; $k_{b2} = 8$; $k_{ub1} = 1$; $k_{ub2} = 1$; $p_1 = 0.1$; $p_3 = 3$; $p_{b1} = 1$; $p_{b2} = 1$; $p_{ub1} = 1$; $p_{ub2} = 1$; $A_{Total} = 8$; $K_{Total} = 4$; $P_{Total} = 3$;

*Figure 2 – figure supplement 1*: Coexistence of multistability and biphasic dose response in various enzymatic models for the same underlying kinetic regime.

 A *Enzyme biphasic - DSP with common kinase and common phosphatase*

- Multi-stability: $k_1 = 100$; $k_2 = 6$; $k_3 = 5$; $k_4 = 20$; $k_{b1} = 0.5$; $k_{b2} = 30$; $k_{b3} = 100$; $k_{b4} = 750$; $k_{ub1} = 1$; $k_{ub2} = 1$; $k_{ub3} = 1$; $k_{ub4} = 1$; $A_{Total} = 65$; $P_{Total} = 30$;
- Enzyme biphasic: $k_1 = 100$; $k_2 = 6$; $k_3 = 5$; $k_4 = 20$; $k_{b1} = 0.5$; $k_{b2} = 30$; $k_{b3} = 100$; $k_{b4} = 750$; $k_{ub1} = 1$; $k_{ub2} = 1$; $k_{ub3} = 1$; $k_{ub4} = 1$; $A_{Total} = 80$; $P_{Total} = 30$;

B *Substrate biphasic - DSP with common kinase and common phosphatase*

$k_1 = 0.1$; $k_2 = 1$; $k_3 = 1$; $k_4 = 1$; $k_{b1} = 10$; $k_{b2} = 1$; $k_{b3} = 0.1$; $k_{b4} = 1$; $k_{ub1} = 1$; $k_{ub2} = 1$; $k_{ub3} = 1$; $k_{ub4} = 1$; $K_{Total} = 2.5$; $P_{Total} = 1$;

C *Substrate biphasic - DSP with common kinase and separate phosphatase*

$k_1 = 0.1$; $k_2 = 1$; $k_3 = 1$; $k_4 = 1$; $k_{b1} = 10$; $k_{b2} = 1$; $k_{b3} = 0.1$; $k_{b4} = 1$; $k_{ub1} = 1$; $k_{ub2} = 1$; $k_{ub3} = 1$; $k_{ub4} = 1$; $K_{Total} = 5$; $P1_{Total} = 1$; $P2_{Total} = 2$;

D *Enzyme biphasic - DSP with separate kinase and common phosphatase*

$k_1 = 30$; $k_2 = 5$; $k_3 = 50$; $k_4 = 1$; $k_{b1} = 25$; $k_{b2} = 10$; $k_{b3} = 10$; $k_{b4} = 7$; $k_{ub1} = 1$; $k_{ub2} = 1$; $k_{ub3} = 1$; $k_{ub4} = 1$; $A_{Total} = 200$; $K1_{Total} = 0.1$; $P_{Total} = 6$;

E Substrate biphasic response - Cascaded enzymatic network

$k_1 = 80$; $= 35$; $= 600$; $= 400$; $= 1$; $= 1$; $= 5$; $= 60$; $= 40$; $= 25$; $= 1$; $= 1$; $= 45$; $= 10$; $= 16$;

*Figure 2—figure supplement 1*: Biphasic responses in Random ordered Double site modification network with separate enzymes effecting each modification.

A *Substrate biphasic in random ordered DSP with separate enzymes*

$k_1 = 1$; $k_2 = 1$; $k_3 = 1$; $k_4 = 1$; $a_1 = 1$; $a_2 = 4$; $a_3 = 1$; $a_4 = 1$; $k_{b1} = 1$; $k_{b2} = 1$; $k_{b3} = 1$; $k_{b4} = 1$; $a_{b1} = 1$; $a_{b2} = 1$; $a_{b3} = 1$; $a_{b4} = 1$; $k_{ub1} = 1$; $k_{ub2} = 1$; $k_{ub3} = 1$; $k_{ub4} = 1$; $a_{ub1} = 1$; $a_{ub2} = 1$; $a_{ub3} = 1$; $a_{ub4} = 1$; $K1_{Total} = 2$; $K2_{Total} = 2$; $P1_{Total} = 3$; $P2_{Total} = 3$;

B Enzyme biphasic in random ordered DSP with separate enzymes

$k_1 = 3$; $k_2 = 1$; $k_3 = 0.1$; $k_4 = 1$; $a_1 = 0.1$; $a_2 = 10$; $a_3 = 1$; $a_4 = 1$; $k_{b1} = 1$; $k_{b2} = 1$; $k_{b3} = 1$; $k_{b4} = 1$; $a_{b1} = 1$; $a_{b2} = 1$; $a_{b3} = 1$; $a_{b4} = 1$; $k_{ub1} = 1$; $k_{ub2} = 1$; $k_{ub3} = 1$; $k_{ub4} = 1$; $a_{ub1} = 1$; $a_{ub2} = 1$; $a_{ub3} = 1$; $a_{ub4} = 1$; $A_{Total} = 50$; $K1_{Total} = 2$; $P1_{Total} = 12$; $P2_{Total} = 12$;

*Figure 2—figure supplement 3* & *Figure 2—figure supplement 4*: Semi-analytical approach for determining the effect of species total amounts in realizing biphasic responses.

The parameters used here have been referenced from literature (as collated by **Witzel and Blüthgen, 2018** in supporting materials document S1). The parameters have dimensions; the binding constants are in the units of $s^{-1}\mu M^{-1}$, while the unbinding and catalytic constants are in the units of $s^{-1}$. The total amounts are in the units of $\mu M$. See **Supplementary file 2** for more details. Note that in addition to specifying intrinsic kinetic parameters, some species total amounts are fixed at specific levels, within physiological ranges. This is to allow for the creation and easy visualization of contour plots. In some cases, a variable is fixed at a particular value. This allows us to determine, semi-analytically, from the requirement of the steady state and the presence of a biphasic peak, what other variables and associated total amounts of species will be. The contour plots for total amounts are plotted in *Figure 2—figure supplement 4*.

A *Coupled covalent modification system (common kinase and phosphatase) - Substrate biphasic*

$k_{b1} = 0.18$; $k_{b2} = 0.18$; $k_{b3} = 0.18$; $k_{b4} = 0.18$; $k_{ub1} = 0.18$; $k_{ub2} = 0.18$; $k_{ub3} = 0.18$; $k_{ub4} = 0.18$; $k_1 = 0.0147$; $k_2 = 0.107$; $k_3 = 0.0385$; $k_4 = 0.0515$; In addition, $P_{total} = 5$ and K=1 was used to obtain curves in the relevant concentration variables corresponding to the onset (peak) of a biphasic response. Every point on the curve corresponded to the peak of a biphasic response for total amounts of species shown in *Figure 2—figure supplement 4*. A similar approach was used for all other plots below.

B *Two-tier cascaded modification system (common phosphatase) - substrate biphasic*

$k_{b1} = 0.18$; $k_{b2} = 0.18$; $k_{b3} = 0.18$; $k_{b4} = 0.18$; $k_{ub1} = 0.18$; $k_{ub2} = 0.18$; $k_{ub3} = 0.18$; $k_{ub4} = 0.18$; $k_1 = 0.1$; $k_2 = 0.01$; $k_3 = 0.01$; $k_4 = 0.1$; In addition $P_{total} = 2$ and K=0.2 was used to obtain

the curves in the concentration variables corresponding to the onset (peak) of a biphasic response.

C *Double site modification system (common kinase and separate phosphatase) - substrate biphasic*

$k_{b1} = 0.18$; $k_{b2} = 0.18$; $k_{b3} = 0.18$; $k_{b4} = 0.18$; $k_{ub1} = 0.18$; $k_{ub2} = 0.18$; $k_{ub3} = 0.18$; $k_{ub4} = 0.18$; $k_1 = 0.0747$; $k_2 = 0.0357$; $k_3 = 0.0585$; $k_4 = 0.0315$; In addition $P_{1,total} = 5$ and $P_{2,total} = 1$ was used to obtain the curves in the concentration variables corresponding to the onset (peak) of a biphasic response.

D *Double site modification system (separate kinase and common phosphatase) - enzyme biphasic*

$k_{b1} = 0.18$; $k_{b2} = 0.18$; $k_{b3} = 0.18$; $k_{b4} = 0.18$; $k_{ub1} = 0.18$; $k_{ub2} = 0.18$; $k_{ub3} = 0.18$; $k_{ub4} = 0.18$; $k_1 = 0.0747$; $k_2 = 0.0357$; $k_3 = 0.0585$; $k_4 = 0.0315$; In addition $P_{total} = 5$ and $K_1 = 0.2$ was used to obtain the curves in the concentration variables corresponding to the onset (peak) of a biphasic response.

E Double site modification system (common kinase and phosphatase) - substrate and enzyme biphasic

$k_{b1} = 0.18$; $k_{b2} = 0.18$; $k_{b3} = 0.18$; $k_{b4} = 0.18$; $k_{ub1} = 0.18$; $k_{ub2} = 0.18$; $k_{ub3} = 0.18$; $k_{ub4} = 0.18$; $k_1 = 0.0747$; $k_2 = 0.0357$; $k_3 = 0.0585$; $k_4 = 0.0315$; In addition $P_{total} = 20$ was used to obtain the curves in the concentration variables corresponding to the onset (peak) of a biphasic response, for the substrate biphasic response. $P_{total} = 15$ was used in the instance of the enzyme biphasic response.

**Figure 3**: Role of biphasic response within interactions in network motif responses.

A *Upstream biphasic regulation of covalent modification cycle*

$u_a = 1$; $u_b = 0.01$; $u_d = 1$; $b_1 = 2.42*4$; $b_2 = \text{sqrt}(2)/20$; $k_1 = 4$; $k_2 = 1$; $k_{b1} = 1$; $k_{b2} = 1$; $k_{ub1} = 1$; $k_{ub2} = 1$; $A_{Total} = 20$; $K_{Total} = 5$; $P_{Total} = 3$;

B *Upstream biphasic regulation of double site modification system (with common enzymes)*

(a) $u_a = 1$; $u_b = 0.01$; $u_d = 1$; $b_1 = 2.42*1$; $b_2 = \text{sqrt}(2)/3$; $k_1 = 1$; $k_2 = 2$; $k_3 = 1$; $k_4 = 1$; $k_{b1} = 1$; $k_{b2} = 1$; $k_{b3} = 1$; $k_{b4} = 1$; $k_{ub1} = 1$; $k_{ub2} = 1$; $k_{ub3} = 1$; $k_{ub4} = 1$; $A_{Total} = 20$; $K_{Total} = 4$; $P_{Total} = 3$;
(b) $u_a = 0.1$; $u_b = 0.01$; $u_d = 0.1$; $b_1 = 2.42*0.1$; $b_2 = \text{sqrt}(2)/8$; $k_1 = 1$; $k_2 = 5$; $k_3 = 1$; $k_4 = 8$; $k_{b1} = 1$; $k_{b2} = 1$; $k_{b3} = 1$; $k_{b4} = 1$; $k_{ub1} = 1$; $k_{ub2} = 1$; $k_{ub3} = 1$; $k_{ub4} = 1$; $A_{Total} = 20$; $K_{Total} = 5$; $P_{Total} = 2$;
(c) $u_a = 0.02$; $u_b = 0.001$; $u_d = 40$; $b_1 = 2.42*0.5$; $b_2 = \text{sqrt}(2)/2$; $k_1 = 0.5$; $k_2 = 0.1$; $k_3 = 10$; $k_4 = 1$; $k_{b1} = 10$; $k_{b2} = 10$; $k_{b3} = 1$; $k_{b4} = 1$; $k_{ub1} = 1$; $k_{ub2} = 1$; $k_{ub3} = 1$; $k_{ub4} = 1$; $A_{Total} = 60$; $K_{Total} = 75$; $P_{Total} = 3$;

C *Incoherent feedforward regulation of double site modification system (with common enzymes)*

(a) $u_a = 0.02$; $u_b = 0.001$; $u_d = 40$; $d_a = 0.1$; $d_b = 0.001$; $d_d = 0.4$; $k_1 = 3$; $k_2 = 0.1$; $k_3 = 10$; $k_4 = 1$; $k_{b1} = 10$; $k_{b2} = 10$; $k_{b3} = 1$; $k_{b4} = 1$; $k_{ub1} = 1$; $k_{ub2} = 1$; $k_{ub3} = 1$; $k_{ub4} = 1$; $A_{Total} = 60$; $K_{Total} = 75$; $P_{Total} = 3$;
(b) $u_a = 0.2$; $u_b = 0.001$; $u_d = 30$; $d_a = 4$; $d_b = 0.001$; $d_d = 20$; $k_1 = 0.5$; $k_2 = 0.1$; $k_3 = 10$; $k_4 = 1$; $k_{b1} = 10$; $k_{b2} = 10$; $k_{b3} = 1$; $k_{b4} = 1$; $k_{ub1} = 1$; $k_{ub2} = 1$; $k_{ub3} = 1$; $k_{ub4} = 1$; $A_{Total} = 100$; $K_{Total} = 75$; $P_{Total} = 2$;

D *Incoherent feedforward network motif*

$b_1 = 2.42*2$; $b_2 = \text{sqrt}(2)/2$; $k_1 = 1$; $k_2 = 1$; $K_1 = 0.1$; $K_2 = 1$; $p_1 = 1$; $p_2 = 1$; $P_1 = 0.1$; $P_2 = 4$; $d_1 = 2$; $d_2 = 2$; $D_1 = 0.1$; $D_2 = 1$; $A_{Total} = 1$; $B_{Total} = 2$; $R_{Total} = 4$;

E *Positive feedback network motif*

$b_1 = 2.42*0.1$; $b_2 = \text{sqrt}(2)/3$; $k_0 = 0.1$; $k_1 = 0.4$; $K_1 = 10$; $k_2 = 2$; $K_2 = 2$; $k_3 = 1$; $k_4 = 1$; $K_4 = 0.1$; $A_{Total} = 3$; $R_{Total} = 10$;

F *Negative feedback network motif*

$b_1 = 13.0266*0.1$; $b_2 = 1$; $k_1 = 0.1$; $K_1 = 2$; $k_2 = 0.1$; $k_3 = 0.1$; $k_4 = 0.1$; $K_4 = 0.5$; $R_{Total} = 50$; $A_{Total} = 5$;

*Figure 3—figure supplement 1*. Effect of biphasic interaction on feedback network motifs (Open systems).

A *Positive feedback motif*

$b_1 = 0.1$; $b_2 = 5$; $k_0 = 0.01$; $k_1 = 0.5$; $k_2 = 1$; $k_3 = 1$; $k_4 = 1$; $k_4 = 0.5$; $K_4 = 0.1$; $A_{Total} = 3$;

B Negative feedback motif

$b_1 = 0.5$, $b_2 = 0.8$; $k_0 = 0.1$; $k_1 = 0.1$; $k_2 = 0.001$; $k_3 = 0.1$; $k_4 = 2$; $K_3 = 0.01$; $K_4 = 0.01$; $A_{Total} = 3$;

*Figure 3—figure supplement 1* Effect of biphasic interaction on integral control motif.

A *Expected system response*

$ka_d = 1$; $kp_a = 1$; $kp_d = 1$; $km_a = 1$; $km_d = 1$;

B *Upstream biphasic regulation by signal*

$b_2 = 0.1$; $b_2 = 0.8$; $b_2 = 2.5$;

C Biphasic in interaction

$b_2 = 0.1$; $b_2 = 0.8$; $b_2 = 2.5$;

*Figure 4*: Presence and consequence of biphasic dose responses in cellular signaling arising from enzyme-mediated substrate modification.

A *Substrate biphasic in the Erk regulation model*

$k_1 = 0.0747$; $k_2 = 0.0957$; $k_4 = 0.0104$; $k_{b1} = 0.18$; $k_{b2} = 0.18$; $k_{b4} = 0.18$; $k_{ub1} = 0.27$; $k_{ub2} = 0.27$; $k_{ub4} = 0.27$; $p_2 = 0.0338$; $p_3 = 0.00851$; $p_4 = 0.0106$; $p_{b2} = 0.18$; $p_{b3} = 0.18$; $p_{b4} = 0.18$; $p_{ub2} = 0.27$; $p_{ub3} = 0.27$; $p_{ub4} = 0.27$; $MeK_{Total} = 0.12$; $P_{Total} = 1$;

B *Enzyme biphasic in the Erk regulation model*

$k_1 = 2$; $k_2 = 1$; $k_4 = 1$; $k_{b1} = 0.18$; $k_{b2} = 0.18$; $k_{b4} = 0.18$; $k_{ub1} = 0.27$; $k_{ub2} = 0.27$; $k_{ub4} = 0.27$; $p_2 = 0.1$; $p_3 = 5$; $p_4 = 0.1$; $p_{b2} = 0.18$; $p_{b3} = 0.18$; $p_{b4} = 0.18$; $p_{ub2} = 0.27$; $p_{ub3} = 0.27$; $p_{ub4} = 0.27$; $Erk_{Total} = 100$; $P_{Total} = 5$;

C Simultaneous presence of substrate and enzyme biphasic response in the Erk regulation model

$k_1 = 0.1$; $k_2 = 1$; $k_4 = 10$; $k_{b1} = 1$; $k_{b2} = 1$; $k_{b4} = 1$; $k_{ub1} = 1$; $k_{ub2} = 1$; $k_{ub4} = 1$; $p_2 = 0.01$; $p_3 = 40$; $p_4 = 1$; $p_{b2} = 1$; $p_{b3} = 1$; $p_{b4} = 1$; $p_{ub2} = 1$; $p_{ub3} = 1$; $p_{ub4} = 1$; $P_{Total} = 5$;

# Acknowledgements

We gratefully acknowledge funding to VR via a Presidential Scholarship at Imperial College London.

# Additional information

### Funding

| Funder | Grant reference number | Author |
| --- | --- | --- |
| Imperial College London | Presidential Scholarship | Vaidhiswaran Ramesh |

The funders had no role in study design, data collection and interpretation, or the decision to submit the work for publication.

## Author contributions
Vaidhiswaran Ramesh, Software, Formal analysis, Investigation, Methodology, Writing – original draft, Writing – review and editing; J Krishnan, Conceptualization, Formal analysis, Supervision, Investigation, Methodology, Writing – original draft, Writing – review and editing

## Author ORCIDs
Vaidhiswaran Ramesh (ID) http://orcid.org/0000-0002-8514-4657
J Krishnan (ID) https://orcid.org/0000-0001-6196-2033

## Decision letter and Author response
Decision letter https://doi.org/10.7554/eLife.86520.sa1
Author response https://doi.org/10.7554/eLife.86520.sa2

---

## Additional files

### Supplementary files
- MDAR checklist
- Supplementary file 1. PDF Version of *Source code 1*.
- Supplementary file 2. PDF Version of *Source code 2*.
- Source code 1. Analysis of models.
- Source code 2. Further analysis of models.

### Data availability
The current manuscript contains results which are computational and analytical (mathematical). These are all presented and discussed in the main text. Maple code was used for establishing the analytical results and this is uploaded as *Source code 1 and 2* (Maple) and in pdf format as supplementary files (*Supplementary files 1 and 2*). Source code 1 (and Supplementary file 1) contains the basic analytical and semi-analytical work, while Source code 2 (and Supplementary file 2) contains additional analysis. The code for generating the computational results has been deposited on GitHub (copy archived at *Ramesh, 2022*).

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
