## [Editor Report]

This study presents a useful mathematical analysis of different signaling networks in an attempt to provide general rules that give rise to biphasic responses, a widely observed behavior in biology in which the outputs of the network depend non-monotonically on the inputs. The methodology is comprehensive and solid, and should provide a useful starting point for systems biologists and quantitative biologists interested in engineering synthetic biological systems and for mechanistically understanding biphasic responses in natural biological systems.

---

## [Decision Letter]

**Decision letter after peer review:**

Thank you for submitting your article "Biphasic responses in cell signalling: A unified approach" for consideration by *eLife*. Your article has been reviewed by 2 peer reviewers, and the evaluation has been overseen by a Reviewing Editor and Aleksandra Walczak as the Senior Editor. The following individual involved in the review of your submission has agreed to reveal their identity: Jane Kondev (Reviewer #1).

Essential revisions:

Overall, both Reviewers expressed difficulty in assessing the generality and importance of the results.

(1a) Presentation of Results: Clarifying how specific modules were chosen and their importance: Reviewer #1 states: "it was hard to assess to what extent such rules exist as behaviors can change depending on the context of a larger network in which the smaller biphasic one is embedded." Relatedly, Reviewer #2 states "It is unclear how why and how the specific biochemical modules were chosen. The Results section reads more like a list of different explorations that were carried out than a systematic progression of investigations leading to specific key insights." It will be important for the authors to improve the way the results are presented, and specifically to clarify/organize the Results section such that the reader follows how the authors discovered key mechanisms through the specific examples they consider.

(1b) Presentation of Results: Importance of the results: Related to the above, both Reviewers commented that the description of the results is formal and limited. For example, Reviewer #2 states "For most of the investigations, the results from investigations are stated but not what the authors learned from those calculations mechanistically." The authors will need to revise their manuscript to address these concerns to ensure their results are not of limited utility.

(2) Choice of Parameters: Both Reviewers raised several, important concerns about the choice of parameters used. These concerns will need to be addressed in a revision, which will help in clarifying the extent to which the results are (a) meaningful biologically and (b) general.

*Reviewer #1 (Recommendations for the authors):*

It wasn't clear which results are general and which are dependent on the kinetic parameters being used. Also, why were the rate parameters chosen to have the specific values used? The parameters are also dimensionless which makes it hard to form any intuition about the results. The same goes for choices for the numbers that specify Atotal, Ktotal, Ptotal. And there are some really "mysterious" choices like b1 = 2.42*1? It would be very helpful if these choices were explained.

*Reviewer #2 (Recommendations for the authors):*

Below are some concerns that diminish enthusiasm and a few suggestions to address those.

1. It is unclear why and how the specific biochemical modules were chosen. The Results section reads more like a list of different explorations that were carried out than a systematic progression of investigations leading to specific key insights. As a result, it becomes difficult to distill the main mechanisms the author learned from these investigations. The titles of the subsections in the Results section are a mixture of topics (e.g., "the modification biochemistry level") or the conclusion of an investigation (e.g., "Multisite substrate modification and commonality of enzyme action promotes biphasic dose responses") which makes it difficult to follow the logic the authors are pursuing to glean the mechanisms. I think reorganizing this section such that the reader follows how the authors discovered the key mechanisms by systematically considering the examples will be helpful.

2. For most of the investigations, the results from investigations are stated but not what the authors learned from those calculations mechanistically. For example, on page 3, the authors state "We observe that the double site modification (DSP) network.. is capable of exhibiting both substrate and enzyme biphasic responses..", and then some conditions given by the rates are provided, however, what did we learn physically/mechanistically from these calculations are not provided. I think, unless such interpretations are given, the explorations remain as standalone examples which would be difficult to use in future modeling or experiments.

3. There is no discussion about how related the ranges of concentrations and rates giving rise to biphasic responses are to real biological systems. I think this is necessary to assess the applicability of the analysis for biological systems. Perhaps the authors could include a table or a figure that relates the examples considered here to different biological systems or synthetic biological systems. The subsection on "Methodology of analysis and testable predictions" in the Discussion section appears to be rather generic. I think if the authors can point to some specific results from their investigations that can be tested in experiments that will strengthen the conclusions of the manuscript.

4. Most of the analyses appear to be carried out in steady states, however, many of the responses in biological systems occur transiently. How would the conclusions drawn here apply to those cases?

---

## [Author Response]

Essential revisions:Overall, both Reviewers expressed difficulty in assessing the generality and importance of the results.(1a) Presentation of Results: Clarifying how specific modules were chosen and their importance: Reviewer #1 states: "it was hard to assess to what extent such rules exist as behaviors can change depending on the context of a larger network in which the smaller biphasic one is embedded." Relatedly, Reviewer #2 states "It is unclear how why and how the specific biochemical modules were chosen. The Results section reads more like a list of different explorations that were carried out than a systematic progression of investigations leading to specific key insights." It will be important for the authors to improve the way the results are presented, and specifically to clarify/organize the Results section such that the reader follows how the authors discovered key mechanisms through the specific examples they consider.

We have improved the presentation in multiple respects. Firstly, we have made the three different sets of studies into three subsections.

Then with respect to the choice of modules, we have inserted/edited the following paragraphs at the beginning of the substrate modification biochemistry level starting on page 3:

“Our analysis begins with the modification biochemistry level.

Biphasic responses at the modification biochemistry level, emerge from competing effects engendered by sequestration of enzymes/substrates (which in turn is simply a consequence of complex formation in any enzymatic biochemical reaction).

We explore a range of biochemical modification systems which bring to light exactly when such sequestration effects will provide a competing effect giving rise to a biphasic response. We start with the simplest modification system (a covalent modification cycle) and increase the complexity systematically (mimicking biologically observed scenarios), allowing for multiple enzymes and/or substrates and sharing of enzymes and substrates. As seen below, these provide the ingredients for generating competing effects leading to biphasic responses. Note that in all these instances, if the sequestration effects are removed (eg. all substrate modification steps in the unsaturated limit), the possibility of a biphasic response is eliminated.

Our analysis at the substrate modification level is underpinned by an exploration of a broad suite of basic systems, which represent different extensions of the covalent modification cycle (the basic unit of reversible substrate modification). Thus, we consider (i) the covalent modification cycle, along with additional interactions (ii) multiple modifications of a substrate, mediated by either common/separate enzymes (iii) enzymatic modifications cascades (where the modified substrate at one stage of the cascade is the enzyme for the next), again considering both common and separate phosphatases. (iv) two different covalent modification cycles whose conversion is mediated by common enzymes (either kinase or phosphatase or both).

Thus, this suite of basic systems allows us to thoroughly explore the impact of sharing of enzymes or substrates, or both, in enabling (or precluding) the possibility of biphasic responses. In particular, the structural requirements for biphasic responses which emerge allow us to pinpoint key drivers and minimum requirements for such responses providing valuable mechanistic insight in the process.”

We then also present a paragraph on the organization of the results in this subsection (p3):

“The organization of the results. We present the results of the analysis of each of these systems, in sequence. In each case (a) we indicate whether or not a biphasic response (of either type) is possible (b) we specify whether there are any restrictions in intrinsic kinetic parameters for this to happen (this is based on analytical work) (c) in systems where biphasic responses are possible, we present the basic mechanistic requirements and intuition emerging from the analysis. (a) and (b) are also depicted in Figure 2: here when biphasic responses are possible, a sample biphasic response is shown. We present a summary of additional parametric analysis for the systems which can exhibit biphasic responses, at the end of this subsection.”

Similarly, in the network level subsection, we also insert a paragraph on the organization of the results: (p7)

“Organization of results. We probe the impact of a biphasic interaction pattern in a network. To do this, we first probe the effect of biphasic regulation by signal of a basic covalent modification cycle (a representative node of a network). We then explore the impact of biphasic regulation internal to networks, by examining ubiquitous network motifs (feedforward and feedback motifs). Finally, we explore the interplay of biphasic regulation.interaction, and biphasic behaviour intrinsic to a node in a network, the latter arising from substrate modification biochemistry.”

Furthermore, in response to Reviewer 2, we have now explained the basis of the different modification systems we consider. Their systematic exploration provides the origins for biphasic responses and the associated mechanistic insights are presented (see below). Similarly at the network level, we start with ubiquitous network motifs with characteristic behaviour and show how a biphasic pattern of interaction in that motif can alter the motif behaviour.

Also, in response to Reviewer 1: we first emphasize that we start at the substrate modification level to uncover drivers of biphasic responses at this level. Biphasic responses arise from an inbuilt competing effect and we demonstrate different ways in which such an inbuilt competing effect arises, through sharing of enzymes or substrates. While it is true that the behaviour can change as part of a network (a) It still remains that there are these in-built competing effects which can generate biphasic responses (both substrate and enzyme) and that can manifest at a pathway or network level (b) the fact that behaviour at a network level may be altered is exactly why we consider studies at the network level showing both biphasic patterns in interaction the overall behaviour is determined by the motif and the biphasic pattern of interaction and studies involving interaction of biphasic responses at both the network and substrate modification level!!

(1b) Presentation of Results: Importance of the results: Related to the above, both Reviewers commented that the description of the results is formal and limited. For example, Reviewer #2 states "For most of the investigations, the results from investigations are stated but not what the authors learned from those calculations mechanistically." The authors will need to revise their manuscript to address these concerns to ensure their results are not of limited utility.

In response to this point we have now presented mechanistic insight into why biphasic responses occur in different instances.

For instance, in the Substrate Modification Level subsection:

(Covalent modification cycle with additional interaction) (p4)

“Examining the two cases above illuminates why an enzyme-biphasic response emerges in the latter case. The extra protein interaction creates a competing effect: while an increase in kinase enzyme concentration, generally favours increased concentrations of the modified substrate, it also has a secondary effect of binding to the modified substrate and sequestering it, thus reducing the (free) modified substrate concentration. This creates the in-built competing effect resulting in a biphasic response.”

(Multisite substrate modification systems): (p5)

“An examination of all these cases reveals the factors responsible for generating the biphasic response (also see Suwanmajo et al., 2013). Firstly the possibility of an enzyme biphasic response can be explained as follows:

in the common-kinase common-phosphatase case, while increasing the kinase concentration favours an increase in the fully modified substrate, it can also have an auxilliary effect: sequestering the partially modified substrate in a kinase complex. This has the effect of making more phosphatase available to dephosphorylate the fully modified substrate and this provides the in-built competing effect. The crucial factor here is the fact that the phosphatase dephosphorylates both fully modified and partially modified substrate. This is further borne out by the fact that even the separate kinase common phosphatase case can result in an enzyme biphasic response when the concentration of the second kinase is varied (for the same reason).

With regard to substrate biphasic responses, we find that the common kinase separate phosphatase case shows this. Here, increasing the total substrate concentration has two effects: the natural effect of increasing all substrate concentrations including that of the fully modified substrate, and another effect: that of reducing the free enzyme concentration. This, when combined with co-operativity, the effect of having two substrates reduce the free kinase concentration (the unmodified and partially modified form) provides a sufficiently strong nonlinear effect, allowing for a competing effect resulting in the biphasic response (confirmed by analytical work). Naturally this effect is present in the common kinase common phosphatase case as well.”

(Two-step enzymatic cascade with shared phosphatase) (p5)

“This latter case can be understood as follows: increasing the total substrate concentration in the second tier has the effect of increasing the sequestering of the modified species in the first tier (in a complex with the unmodified species of the second tier). This has the consequence of making more phosphatase available, which can then dephosphorylate the modified species in the second tier. This competing effect (reminiscent of that responsible for an enzyme biphasic in multisite substrate modification above) allows for the manifestation of the substrate biphasic response.”

(Coupled covalent modification cycles): (p6)

“The fact that covalent modification cycles involving a shared kinase and a shared phosphatase can result in a substrate biphasic response (for one of the substrates) can be understood as follows. Increasing one of the substrates can under certain situations result in a combination of effects: one the reduction of free kinase (as seen previously) and the other an increase free phosphatase (since the reduction in free kinase implies a lower level of modification of the other substrate, and this implies less phosphatase used there). Both these effects have been seen in different guises previously, but here their combined co-operative effect provides the nonlinearity to create the competing effect.”

A similar discussion is inserted in the subsection on the network level: (p7 penultimate paragraph)

“The basic intuition in both instances is that the biphasic interaction pattern causes deviation from expected behaviour, simply because in a certain range, the nature of regulation changes from positive to negative. This negative regulation can combine with an overall negative feedback regulation pattern to give behaviour reminiscent of a positive feedback (eg. multistability).”

Finally in the subsection on ERK: (p9, second paragraph)

“All in all, we can conclude that the commonality of kinase in different modification steps and the commonality of phosphatase in demodication steps provides the (different) in-built competing effects for enzyme biphasic and substrate biphasic responses to be realized.”

For both these sets of points (1a and 1b), additional paragraphs are included in the conclusions to recapitulate the key points (p9 and p10)*.*

(2) Choice of Parameters: Both Reviewers raised several, important concerns about the choice of parameters used. These concerns will need to be addressed in a revision, which will help in clarifying the extent to which the results are (a) meaningful biologically and (b) general.

These points are addressed in detail and are summarized below:

We first focus on the substrate modification level and highlight the different classes of parameters and their role in the results:

In the text, we mention:

(p3, second paragraph)

“As part of our analysis of biphasic responses, we aimed to assess the extent to which underlying parameters can prevent or enable the presence of biphasic responses (to either type of dose). We first note that the parameters are of two types: intrinsic kinetic rate constants (the majority of the parameters) and the total amounts of enzymes(s) and/or substrate (other than the dose). The main challenge which arises is the fact that there are a number of intrinsic kinetic parameters (which do not cluster into a small number of groups of parameters, like in many physical systems).

From our analysis, we find, especially in studies at the biochemical modification level, three kinds of scenarios: (a) Biphasic responses are impossible for any values of the intrinsic kinetic parameters, thus categorically ruling out their possibility, from structural considerations. (b) Biphasic responses are possible for certain regions of intrinsic kinetic parameter space, which are explicitly characterized (these are necessary conditions). In these cases, we can guarantee the presence of a biphasic response, for suitable values (and ranges) of total amounts of enzymes and substrates. Thus, here, there is a partial restriction on the intrinsic kinetic parameter space for enabling the possibility of a biphasic response. (c) Biphasic responses are possible irrespective of the intrinsic kinetic parameter space. Thus, intrinsic kinetic parameters play no role in restricting the possibility of a biphasic response (and thus biphasic responses are a widespread occurrence in the space of intrinsic kinetic parameters). Furthermore, a biphasic response can be guaranteed for suitable ranges of enzyme and substrate total concentrations.”

We point out that for substrate modification systems, we completely characterize the essential impact of the intrinsic kinetic parameters (in some cases biphasic behaviour is precluded irrespective of their value, in other cases, there are regions of parameter space which are explicitly delineated, where biphasic responses are possible, for suitable levels of total amounts, in other cases, biphasic responses are possible for all values of intrinsic kinetic parameters, for suitable choices of species total amounts). This is also depicted in Figure 2

The only remaining aspects to discuss are the values of species total amounts, as well as whether biphasic responses can be obtained in reasonable parameter regimes (for systems and synthetic biology).

We address the latter point first. First in the cases where the intrinsic kinetic rate constants place a partial restriction on whether or biphasic responses can be obtained, we note that the region (explicitly delineated) in the parameter space of intrinsic constants is a vast region of parameter space, determined by an inequality associated with ratios of catalytic constants. Thus such an equation does not make any restriction on absolute values of rate constants, or set any scale. It is evident from the nature of the inequality that reasonable kinetic parameters can satisfy this inequality.

With regard to total amounts we perform some further analytical as well as semi-analytical work to determine any restrictions or insights into total amounts of species which might facilitate biphasic responses. This provides a tool for determining species total amounts for yielding biphasic responses given intrinsic rate constants (partially or completely).

We have then use parameters close to those (in the same range and in many cases identical) to those studied in Wistel et al. 2018, which in turn is based on experimental values in the ERK system. We find that for every system considered, with kinetic parameters in this range, we can obtain reasonable ranges of species amounts (for this analysis we use dimensional parameters). This is depicted in Figure 2—figure supplement 3 and Figure 2—figure supplement 4, with further details in the Maple document: supplementary file 2 (we do not discuss the CMC with additional interaction, as this has been discussed in the literature previously, Varusai et al. 2015).

Furthermore using a continuity argument we can assert that there are neighbourhoods of these parameter sets where likewise biphasic responses will be obtained with reasonable range of species total amounts

We have included a paragraph in the substrate modification biochemistry level on Further parametric analysis (p6)

“Further parametric analysis.

The analysis above for all systems has demonstrated either the presence or absence of biphasic responses, and analyzed to what extent the intrinsic kinetic parameters provide a fundamental restriction to the possibility of biphasic responses occurring. We now build on that to provide further insights. 1. We first note that when a biphasic response occurs, the role of intrinsic kinetic parameters are either (a) they provide no essential restriction by themselves or (b) there is a region of parameter space, explicitly delineated, which is the region where a biphasic response is possible. 2. In the latter case, in all instances, there is a simple inequality involving relative magnitudes of the ratio of phosphorylation to dephosphorylation rate constants in different modifications which determines whether biphasic responses are possible or not. Equivalently, the inequality can be written in terms of relative magnitudes of products of specific catalytic constants. This emerges from detailed analysis and reflects the fact that when such a condition is satisfied, the possibility of an in-built competing effect sufficiently strong to be capable of giving rise to a biphasic response exists. 3. The type of this inequality is such that multiplying all catalytic constants by a specific constant does not affect the inequality. Thus, this inequality does not set a restriction on the absolute levels of these rate constants. Furthermore, it is clear from the nature of the inequality, that experimentally feasible ranges of kinetic parameters (catalytic constants) can satisfy such an inequality and that this inequality represents a broad range of parameter space.4. We now examine the effect of other parameters (total amounts of enzymes or substrate which are not part of the dose). By proceeding with further analytical and computational work, we can, in some cases, show further restrictions of enzyme or substrate amounts to realize a biphasic response. For instance, we show (see supplementary file 2 for the points below) that higher phosphatase amounts favours enzyme biphasic responses in the DSP common kinase/phosphatase system, and also substrate biphasic in the two tier cascade studied above; a higher amount of phosphatase in the first demodification step favours a substrate biphasic response in common kinase separate phosphatase case. 5. Similarly we show that in both the DSP common kinase/common phosphatase (enzyme biphasic) and in the coupled covalent modification cycles (substrate biphasic), there is a minimum level of concentration of the modified substrate to allow for a biphasic response (this incidentally also places a lower bound on the total substrate concentration for this to happen). In the coupled covalent modification cycle we can also determine which cycle will exhibit the biphasic response. 6. Further parametric exploration into the effects of total amounts and intrinsic kinetic parameters

was performed semi-analytically, starting with the analytical equations and specifying some parameters. This then provides equations for the remaining parameters and total amounts for a biphasic response to be obtained, and this was explored for all models. As part of this analysis, we also completely fixed intrinsic kinetic parameters in ranges obtained experimentally and studied computationally in Witzel2018 and then used semi-analytical approaches to determine the sets of two total amounts of species (enzymes or substrates) where biphasic responses are obtained (one of the set of three total amounts is the dose, this leaves, two other total amounts in the most basic case where there is a single kinase and a single phosphatase: in the case of multiples kinase and/or multiple phosphatases, there are more total amounts, and these total amounts could be explored together, or by having some of them fixed at particular values).

7. We performed semi-analytical and computational explorations for parameters sets for different modifications in the range of those experimentally determined in the MEK system, and explored in Witzel2018: this is summarized in Figure 2—figure supplement 3 and Figure 2—figure supplement 4. We find that biphasic responses can be obtained in ranges of intrinsic kinetic parameters, in a very similar range as that obtained there. In each model we were able to obtain biphasic responses in reasonable physiological ranges of total amounts (and for total amounts which could be employed in synthetic settings). We also explored a second parameter set where the unbinding rate constants were set to zero, and again biphasic responses were obtained in reasonable ranges (results not shown). 8. Building on these semi-analytical and computational analyses and using continuity arguments, we can see that biphasic response can be readily obtained in neighbouring regions in parameter space (both intrinsic kinetic parameters and total amounts). All in all, this complements our analysis of the intrinsic kinetic rate constants in restricting the possibility of biphasic responses (presented earlier), providing additional insights and parametric regions for biphasic responses and how they may be explored further.”

At the network level we simply use network motifs, with parameters giving rise to their representative behaviour (already well studied) and study how a biphasic interaction pattern alters the motif behaviours.

Finally in the case of ERK signalling, we demonstrate that a substrate biphasic can be obtained irrespective of intrinsic kinetic rate constants. In the case of enzyme biphasic, there is a restriction, but from semi-analytical and numerical work, we see that it can readily be obtained in reasonable kinetic parameters. We mention that Wistel et al. in 2018 have also shown substrate biphasic responses for reasonable ranges of parameters, but we go a step further here and show that this can be obtained for all values of intrinsic kinetic rate constants.

Some of these aspects are summarized with additional paragraphs in the Conclusion (p 9) and Models and Methods (p18, Additional approaches).

Taken together this addresses the point about parameters.

Reviewer #1 (Recommendations for the authors):It wasn't clear which results are general and which are dependent on the kinetic parameters being used. Also, why were the rate parameters chosen to have the specific values used? The parameters are also dimensionless which makes it hard to form any intuition about the results. The same goes for choices for the numbers that specify Atotal, Ktotal, Ptotal. And there are some really "mysterious" choices like b1 = 2.42*1? It would be very helpful if these choices were explained.

This is discussed in detail in the paper and in the comments above as well as the Response to Essential Revisions.

The parameters are dimensionless. However, to show that one can obtain biphasic responses in reasonable parameter ranges we have now, in the revision also used dimensional parameters as well, obtained from experiments (Figure 2—figure supplement 3 and Figure 2—figure supplement 4). In many cases in simulations (eg Figure 2), when biphasic responses were shown, they were pictorially demonstrated for a sample set of parameter values. Note that we have analytically characterized the role of intrinsic kinetic parameters, and the plots show examples of biphasic responses in the admissible parameter regimes. In the network analysis, we simply choose parameters so that the motifs exhibit their characteristic behaviour, and make conclusions about the impact of biphasic patterns of interaction. Similarly, in the ERK case, we show how enzyme biphasic and substrate biphasic responses can occur together, and that substrate biphasic responses can occur for any values of intrinsic kinetic parameter values, thus expanding on the results of Wistel et al. 2018.

We have also explained the origin of the parameter in the biphasic pattern of interaction analysis in the Maple document (see p 18 and associated reference to the original Maple document: supplementary file 1, section 5). This is simply to ensure a certain parity between a biphasic response and a monophasic response over a particular range. This is discussed on p111 of the original Maple document: supplementary file 1, equation 2.1 under subsection Choice of parameters for biphasic in interaction.

Reviewer #2 (Recommendations for the authors):Below are some concerns that diminish enthusiasm and a few suggestions to address those.1. It is unclear why and how the specific biochemical modules were chosen. The Results section reads more like a list of different explorations that were carried out than a systematic progression of investigations leading to specific key insights. As a result, it becomes difficult to distill the main mechanisms the author learned from these investigations. The titles of the subsections in the Results section are a mixture of topics (e.g., "the modification biochemistry level") or the conclusion of an investigation (e.g., "Multisite substrate modification and commonality of enzyme action promotes biphasic dose responses") which makes it difficult to follow the logic the authors are pursuing to glean the mechanisms. I think reorganizing this section such that the reader follows how the authors discovered the key mechanisms by systematically considering the examples will be helpful.

This is now discussed in detail. At the substrate modification level, enzyme/substrate sharing represents a key driver of inbuilt competition generating biphasic response, which in turn arises from sharing of enzymes/substrates. We use a suite a biochemical systems which represent systematic augmentations of a covalent modification cycle to explore under which conditions the inbuilt competing effects resulting in biphasic responses may arise, by examining different augmentations of a covalent modification cycle (multisite substrate modification, enzymatic cascades, modification cycles with shared enzymes, modification cycles with additional interactions).

This is discussed in detail in the Response to Essential Revisions

We explicitly discuss the choice of systems we focus on here. Biphasic responses arise from an in-built competing effect which arises from enzyme/substrate sharing. The suite of models which we examine at the substrate modification level, represents systematic expansions of a basic covalent modification cycle in different ways, and this allows us to explore and pin down when exactly such a competing effect giving rise to a biphasic response arises.

The mechanistic underpinning of the results is also now explicitly discussed (see response to next point).

Finally, we have included paragraphs early in both the subsection on substrate modification and the network level, explaining the organization of the results (p3, p7)

We have denoted the three sets of investigations: substrate modification level, network level and ERK as separate subsections in the revision and agree that by listing them as paragraph headings slightly obscures the organization and flow of the results.

2. For most of the investigations, the results from investigations are stated but not what the authors learned from those calculations mechanistically. For example, on page 3, the authors state "We observe that the double site modification (DSP) network.. is capable of exhibiting both substrate and enzyme biphasic responses..", and then some conditions given by the rates are provided, however, what did we learn physically/mechanistically from these calculations are not provided. I think, unless such interpretations are given, the explorations remain as standalone examples which would be difficult to use in future modeling or experiments.

This point is well taken. We have now included a number of paragraphs especially in the substrate modification level subsection, but also the network level and ERK to explain the intuition and the mechanistic underpinning behind the results.

In particular every substrate building block system exhibiting biphasic responses is now accompanied with some commentary related to the mechanistic underpinning and the underlying intuition.

Complementing this is also a more explicit and expanded discussion of parameters.

This we believe will help a reader use further in both modelling and experiments.

These specific additions are discussed in detail in the Response to Essential Revisions.

3. There is no discussion about how related the ranges of concentrations and rates giving rise to biphasic responses are to real biological systems. I think this is necessary to assess the applicability of the analysis for biological systems. Perhaps the authors could include a table or a figure that relates the examples considered here to different biological systems or synthetic biological systems. The subsection on "Methodology of analysis and testable predictions" in the Discussion section appears to be rather generic. I think if the authors can point to some specific results from their investigations that can be tested in experiments that will strengthen the conclusions of the manuscript.

With respect to rates and constants, first as mentioned above, especially in the substrate modification level and also ERK, we characterize as far as possible the impact of intrinsic rate constants. We show how in some instances biphasic responses are impossible, in other cases always possible (irrespective of intrinsic rate constants, for suitable total amount of species) and in other cases are obtained in broad ranges of intrinsic kinetic parameter space.

This incidentally provides relevant information to show that biphasic responses can be obtained in principle in biologically realistic regions of kinetic parameter space. Secondly, we further analyze the role of species amounts in facilitating biphasic responses (see supplementary file 2). Finally, we also perform further analysis using parameters in ranges very close to those seen experimentally, and show that biphasic responses can be obtained for feasible concentrations and total amounts. These points are discussed in detail in Response to Essential Revisions.

All in all, especially in building block systems, biphasic responses when possible, are possible in vast swathes of (intrinsic kinetic) parameter space, and this readily includes realistic ranges of parameters in systems and synthetic biology. The restrictions, when they do occur, are only on relative values of ratios of catalytic rate constants and as such do not set any scale or absolute level of concentrations of species. These points are discussed in detail in a paragraph: Additional Parametric Analysis, at the end of the substrate modification level subsection(p6). Our conclusions at the network level show a sampling of basic results demonstrating how a biphasic interaction pattern can alter system behaviour. The parameters are simply chosen so that the basic motifs—ubiquitous themselves—exhibit their characteristic behaviour which is already well studied in the literature.

We have also expanded on the paragraph on Testable Predictions. We now write (adding to the paragraph):

“The consequences of biphasic regulation in networks could be tested using networks containing biphasic regulation, which could be engineered (and tuned) synthetically. Our analysis also makes a number of conclusions regarding the possibility of different types of biphasic responses in different building blocks of substrate modification. In each of these instances, the relevant building block can either be isolated from conrete cellular pathways, or built synthetically.

Our analysis already predicts which types of systems are capable of biphasic responses (to variation of enzyme or substrate). For the systems where intrinsic kinetic parameters play no role in obstructing the possibility of biphasic responses, our analysis and its extensions (semi-analytical and numerical) can predict that a biphasic response can be obtained and also how substrate/enzyme total amounts must be varied to obtain a biphasic response: this latter prediction requires a knowledge of the intrinsic kinetic rate constants.

For cases where the possibility of biphasic responses occurs in a subset of the parameter space (of intrinsic rate constants), our analysis predicts whether or not a biphasic response is possible (this requires a knowledge of intrinsic rate constants) and for cases where it is possible, cane be used to determine total amounts of substrate or enzyme where that is possible.

Furthermore, we can also make predictions which are independent of parameters.

consider cases where there is a parametric restriction in obtaining biphasic responses: say enzyme (kinase) biphasic responses in the DSP system with common kinase and common phosphatase. We can predict that if an enzyme biphasic is obtained (i.e. for the concentration of the fully modified form as total amount of kinase is varied), then a biphasic response in the concentration of the unmodified form as phosphatase total amount is varied is possible (for the same set of intrinsic kinetic parameters). This is because a parametric condition which allows for a biphasic response in one direction (forward) say, is exactly what allows it in the opposite direction. This emerges from analyzing the relevant parametric expression analysis of an enzyme biphasic in the reverse direction, amounts to analyzing the same system, with altered labels.”

4. Most of the analyses appear to be carried out in steady states, however, many of the responses in biological systems occur transiently. How would the conclusions drawn here apply to those cases?

The analysis here applies at steady state. It applies both to biological processes at steady state, as well as processes where the individual subsystem is essentially at or close to a steady state. This applies to cases where there is a separation of time scales allowing for this.

We have inserted a paragraph in the conclusion related to this.

“All our analysis of systems has focussed on steady states. The implicit assumption here is that for various processes of interest, the systems under consideration quickly approach the steady state.

Therefore, even though other processes of interest may occur transiently, a steady state analysis of these sub-systems provides useful insight.”